# Toward Conservative Planning from Human-AI Preferences in Reinforcement Learning

**Huazhong Wang**
University of California, Irvine
`huazhonw@uci.edu`

**Wenzhuo Zhou**[*]
University of California, Irvine
`wenzhuz3@uci.edu`

## Abstract

We study reinforcement learning (RL) with trajectory preferences, where the RL agent does not receive explicit rewards at each step but instead receives human-AI preferences over pairs of trajectories. Despite growing interest in preference-based reinforcement learning (PbRL), contemporary works cannot robustly learn policies in offline settings with poor data coverage and often lack algorithmic tractability. We propose a novel **M**odel-based **C**onservative **P**lanning (MCP) algorithm for offline PbRL, which leverages a general function class and uses a tractable conservative learning framework to improve the policy upon an arbitrary reference policy. We prove that, MCP can compete with the best policy within data coverage when the reference policy is supported by the data. To the best of our knowledge, MCP is the first provably sample-efficient and computationally tractable offline PbRL algorithm under partial data coverage, without requiring known transition dynamics. We further demonstrate that, with certain structural properties in PbRL dynamics, our algorithm can effectively exploit these structures to relax the partial data coverage requirement and improve regret guarantees. We evaluate MCP on a comprehensive suite of human-in-the-loop benchmarks in Meta-World. Experimental results show that our algorithm achieves competitive performance compared to state-of-the-art offline PbRL algorithms. Our code is provided at `https://github.com/Rshias/MCP`.

## 1 Introduction

Reinforcement learning (RL) has become a prominent framework for addressing sequential decision-making problems. Most of the existing RL algorithms typically assume access to a well-defined reward function that guides policy optimization. However, in many practical applications, the design of an appropriate reward function poses a significant challenge. This difficulty arises from the complexity of accurately capturing human intent, the susceptibility to reward hacking, and the risk of inducing unintended behaviors due to mis-specified objectives [Wirth et al., 2017]. To tackle the problems, the framework of the preference-based reinforcement learning (PbRL) has emerged as a possible solution. Rather than relying on numerical reward signals, PbRL leverages relative preferences obtained from human experts or large language models, thereby avoiding explicitly modeling the reward function [Christiano et al., 2017]. This framework has proven particularly effective in various domains, including games [MacGlashan et al., 2017; Christiano et al., 2017; Warnell et al., 2018], large language models [Ziegler et al., 2019; Stiennon et al., 2020; Wu et al., 2021; Nakano et al., 2021; Ouyang et al., 2022; Glaese et al., 2022; Bai et al., 2022; Ramamurthy et al., 2022; Liu et al., 2023], and robotics [Brown et al., 2019; Shin et al., 2023].

Many existing PbRL algorithms rely on online interaction with the environment, raising concerns regarding sample efficiency and safety [Christiano et al., 2017; Levine et al., 2020]. In contrast, offline PbRL operates on pre-collected datasets annotated with preference information, providing a more practical solution in scenarios where environment interaction is limited or costly [Shin et al., 2023; An et al., 2023; Kim et al., 2023; Hejna & Sadigh, 2023; Choi et al., 2024]. Despite its potential, much of the prior work in offline PbRL requires that the offline data be fully explored and often lacks sample-efficient guarantees [Zhan et al., 2023a]. The situation becomes even more challenging in

---

[*]Corresponding Author.

scenarios where the offline data distribution only partially covers the trajectory distributions induced by some (but not all) comparator policies—that is, under partial coverage [Jin et al., 2021; Cheng et al., 2022]. In such cases, PbRL algorithms generally fail to learn a good policy with near-optimal regret in a polynomial sample complexity.

Recent efforts have sought to address the above-mentioned challenges under function approximation settings. Zhu et al. [2023] proposes a principled algorithm with sample complexity guarantees under partial coverage; however, their algorithm and theoretical guarantee are restricted to linear reward models. Zhan et al. [2023a]; Pace et al. [2024] extends this framework to the general function approximation setting by explicitly modeling the confidence sets and performing conservative learning. Nevertheless, constructing and optimizing over such confidence sets makes their algorithm computationally intractable in practice. Most recently, a concurrent work [Kang & Oh, 2025] introduces a sample-efficient approach which is able to be implemented in practice. Their algorithms either assume known transition dynamics or require fitting an extra value function that depends on the learned transition model via Bellman recursion to perform conservatism. This allows the value function to locally smooth over gaps in data coverage—if the value function is well-approximated. However, this smoothing-based mitigation strategy intrinsically requires the realizability condition on the value function and does not guarantee near-optimal regret under partial coverage [Yu et al., 2020a]. We refer the readers to Table 1 for detailed comparisons to prior works.

Table 1: Comparison of MCP and its two variants with existing methods in terms of data coverage assumptions, additional structural properties, probably approximately correct (PAC) guarantees with polynomial sample complexity, and computational tractability for practical implementation. MCP refers to our algorithm developed for general function approximation. The superscript ★ indicates that the partial coverage condition is effectively refined by MCP through additional structural properties of the dynamics, resulting in improved regret bounds.

| Methods | Partial Coverage | Additional Structure | PAC Guarantee | Tractable Implementation |
|---|---|---|---|---|
| Principled-RLHF [Zhu et al., 2023] | ✓ | Linear function approximation | ✓ | ✓ |
| FREEHAND [Zhan et al., 2023a] | ✓ | - | ✓ | ✗ |
| OPRL [Shin et al., 2023] | ✗ | - | ✗ | ✓ |
| Sim-OPRL [Pace et al., 2024] | ✓ | - | ✓ | ✗ |
| APPO [Kang & Oh, 2025] | ✓ | Known transition dynamics/value function realizability | ✓ | ✓ |
| IPL [Hejna & Sadigh, 2023] | ✗ | - | ✗ | ✓ |
| **MCP** | ✓ | - | ✓ | ✓ |
| **MCP-Factored**★ | ✓ | Factored model | ✓ | ✓ |
| **MCP-KNR**★ | ✓ | Kernelized nonlinear regulators | ✓ | ✓ |

Motivated by the aforementioned challenges, we study the problem from a model-based learning perspective and propose an implicit way for encoding conservatism that is compatible with PbRL under general function approximation, without requiring known transition dynamics. Specifically, we relax the stringent full coverage assumption and instead assume that the offline data only needs to cover the trajectory distribution induced by the (optimal) comparator policy. We introduce MCP: **M**odel-based **C**onservative **P**lanning, which learns a policy through a tractable conservative learning and model-based planning procedure. The resulting policy matches the performance of any comparator policy that is covered by the offline data. MCP addresses the intractability inherent in existing methods that rely on explicitly constructing confidence sets to encode conservatism, and it avoids additional value function modeling by leveraging a purely model-based planning approach. To the best of our knowledge, MCP is the first offline PbRL algorithm that is both provably sample-efficient and computationally tractable under partial data coverage, without assuming access to the true transition dynamics. When instantiated within specialized PbRL structures, MCP further refines the concentrability coefficient in a tighter manner, leading to improved regret guarantees. In environments with factored dynamics, the regret bound of MCP scales with the number of factors and the size of their parent sets, thereby avoiding the exponential dependence on the state dimension present in non-factored models. Moreover, our analysis shows that the regret bound is adaptive to the offline data distribution and remains valid even when the state space is infinite when MCP is applied in kernelized nonlinear regulators (KNRs). Experimentally, we find that MCP achieves competitive performance compared to state-of-the-art offline PbRL methods, using real human feedback across 8 different Meta-World tasks [Yu et al., 2020b]. In addition, the conducted ablation studies demonstrate the robustness and sample efficiency of the proposed algorithm.

## 2 PRELIMINARIES

**Markov Decision Processes.** We consider a finite-horizon time-inhomogeneous Markov Decision Processes(MDP), denoted as a tuple $M = (\mathcal{S}, \mathcal{A}, r, \{P_h\}_{h=0}^{H-1}, H, s_0)$. Here $\mathcal{S}$ represents the state space, and $\mathcal{A}$ denotes the action space. $P_h : \mathcal{S} \times \mathcal{A} \to \Delta(\mathcal{S})$ is the transition dynamics at time step $h$, where $\Delta(\mathcal{S})$ denotes the set of probability distributions over states. $r : \mathscr{T} \to [0, R_{\max}]$ is the reward function, where $\mathscr{T} = (\mathcal{S} \times \mathcal{A})^H$ represents the set of all trajectories of horizon length $H$. $s_0$ is the initial state. We use $r^\star$ to denote the ground-truth reward function, $\{P_h^\star\}_{h=0}^{H-1}$ to denote the ground-truth transition dynamics. The agent follows a history dependent policy $\pi := \{\pi_h\}_{h=0}^{H-1}$, where each $\pi_h : (\mathcal{S} \times \mathcal{A})^{h-1} \times \mathcal{S} \to \Delta(\mathcal{A})$, specifies a distribution over actions at step $h \in [0 : H - 1]$, conditioned on the entire past trajectory. Let $\Pi$ denote the set of all such history-dependent policies. Given a generic reward function $r$ and transition dynamics $P = \{P_h\}_{h=0}^{H-1}$, the expected cumulative reward is defined as $J(\pi; r, P) := \mathbb{E}_{d_P^\pi}[r(\tau)]$, where $d_P^\pi$ denotes the distribution over trajectories induced by executing policy $\pi$ under transition model $P$. We denote $\mathbb{E}_{d_P^\pi}[r(\tau)|s^0]$ as the expected return of trajectories starting from initial state $s^0$. The state-action visitation distribution at step $h$ is defined as: $d_h^\pi(s, a) = \mathbb{P}_P^\pi(s_h = s, a_h = a), \forall h \in [0 : H - 1]$, where $\mathbb{P}_P^\pi$ represents the probability distribution over trajectories generated by executing policy $\pi$ under transition model $P$. Additionally, we denote the trajectory-level distribution under the ground-truth model $P^\star$ as $d^\pi(\tau)$.

**Offline PbRL.** Offline PbRL is a variant of reinforcement learning that deals with situations where the reward function is not directly available and instead must be inferred from human preferences over trajectory pairs. Offline PbRL focuses on learning from a dataset of trajectory pairs $\mathcal{D} = \{(\tau^{n,0}, \tau^{n,1}, y^n)\}_{n=1}^N$, which contains i.i.d trajectory pairs $\tau^{n,0} = \{s_h^{n,0}, a_h^{n,0}\}_{h=0}^{H-1}, \tau^{n,1} = \{s_h^{n,1}, a_h^{n,1}\}_{h=0}^{H-1}$ sampled from reference policy $\mu$ and binary labels $y^n$. Given a pair of trajectories $(\tau^0, \tau^1)$, a human annotator provides a binary preference label $y \in \{0, 1\}$, where $y = 1$ indicates that trajectory $\tau^1$ is preferred over $\tau^0$, and $y = 0$ indicates the opposite.

To model the preference feedback between trajectories, we introduce a link function $\Phi : \mathbb{R} \to [0, 1]$, which is a monotonically increasing function. Given a pair of trajectories $(\tau^0, \tau^1)$, the preference model assumes that the probability of preferring trajectory $\tau^1$ over $\tau^0$ is given by:

$$\mathbb{P}(y = 1|\tau^0, \tau^1) = \mathbb{P}(\tau^1 \text{ preferred over } \tau^0) = \Phi(r^\star(\tau^1) - r^\star(\tau^0))$$

One of the most commonly adopted link functions is the sigmoid function $\sigma(x) = (1 + \exp(-x))^{-1}$. This link function is associated with the Bradley-Terry-Luce (BTL) model [Bradley & Terry, 1952], which effectively models the relative preference between trajectories. And we define $\kappa := (\inf_{x \in [-R_{\max}, R_{\max}]} \Phi'(x))^{-1}$ to measure the non-linearity of the link function $\Phi$. The goal of offline PbRL is to learn a high-performing policy $\pi^{\text{ALG}} \in \Pi$ which satisfies the following guarantee: $J(\pi; r^\star, P^\star) - J(\pi^{\text{ALG}}; r^\star, P^\star) \leq \epsilon$. Here, $\pi$ denotes a comparator policy that the learned policy aims to match or surpass in performance. For the remainder of the paper, we abuse notation by referring to $J(\pi) - J(\pi^{\text{ALG}})$ as the regret.

**General Function Approximation.** In this work, we consider general function approximation for offline PbRL. Specifically, we model the reward and transition dynamics using a family of transition function classes $\{\mathcal{P}_h\}_{h=0}^{H-1}$ and a reward function class $\mathcal{R}$. These classes are expressive enough to capture complex dynamics and reward structures through the use of linear approximators or neural networks. To quantify the complexity of the transition and reward model classes, we adopt the $1/N$-bracketing number metric, denoted as $\mathcal{N}_\mathcal{P}(1/N)$ and $\mathcal{N}_\mathcal{R}(1/N)$, respectively [Geer, 2000].

## 3 PREFERENCE-GUIDED CONSERVATIVE PLANNING

In this section, we present a novel sample-efficient and computationally tractable offline PbRL algorithm, **M**odel-based **C**onservative **P**lanning (MCP). The developed algorithm integrates model-based planning with implicitly encoded conservatism to guarantee the learned policy can compete with any (best) comparator polices within data coverage.

## 3.1 Algorithm Formulation

We begin by presenting the motivation that underpins the design of our algorithm. The major challenge in offline PbRL is that directly performing policy learning based on the learned reward and transition models for agreement with preference feedback is inaccurate and may result in overestimation issues. To get rid of this problem, the existing works heavily rely on constructing explicit confidence sets to perform conservative learning, which is often not computationally tractable. This motivates us to develop an algorithm that alternates between encoding the conservatism into the learned reward and transition models and learning the policy via a model-based planning procedure upon the worst-case models that remain consistent with the observed preferences in offline data.

In MCP, we formulate the main objective to identify a policy $\pi$ that performs favorably relative to a reference distribution $\mu_{ref}$, a common choice is the distribution to induce offline data. Specifically, we aim to maximize the performance difference between the candidate policy and the reference distribution. This relative performance evaluation encourages policy improvement upon the reference policy while avoiding reliance on potentially inaccurate absolute value estimates. Moreover, the evaluation can be easily performed through a model-based planning procedure, and avoids extra value function modeling.

$$\max_{\pi} J(\pi; r, \{P_h\}_{h=0}^{H-1}) - \mathbb{E}_{\tau \sim \mu_{ref}}[r(\tau)].$$

MCP then takes two realizable hypothesis classes for the reward and transition kernels—i.e., $r^\star \in \mathcal{R}$ and $P_h^\star \in \mathcal{P}_h$ for all $h \in [0 : H-1]$—which consist of potential data-consistent candidate models as input, and computes the maximum likelihood models $\widehat{r}$ and $\widehat{P}_h$ using the given offline dataset $\mathcal{D}$. It then formulates a minimax objective function.

$$\max_{\pi \in \Pi} \min_{r \in \mathcal{R}, \{P_h \in \mathcal{P}_h\}_{h=0}^{H-1}} J(\pi; r, \{P_h\}_{h=0}^{H-1}) - \mathbb{E}_{\tau \sim \mu_{ref}}[r(\tau)] + \lambda_1 \mathcal{E}_1(r; \mathcal{D}) + \lambda_2 \mathcal{E}_2(\{P_h\}_{h=0}^{H-1}; \mathcal{D}),$$
(3.1)

where the conservatism is implicitly encoded via regularizing the empirical absolute discrepancy between the learning targets, i.e., $r$ and $P_h$, and the data-consistent models $\widehat{r}$ and $\widehat{P}_h$:

$$\mathcal{E}_1(r; \mathcal{D}) = \frac{1}{N} \sum_{n=1}^{N} \left\| \left(r(\tau^{n,1}) - r(\tau^{n,0})\right) - \left(\widehat{r}(\tau^{n,1}) - \widehat{r}(\tau^{n,0})\right) \right\|,$$

$$\mathcal{E}_2(\{P_h\}_{h=0}^{H-1}; \mathcal{D}) = \frac{1}{N} \sum_{n=1}^{N} \sum_{h=0}^{H-1} \sum_{i=0}^{1} \left\| P_h(s_{h+1}^{n,i} | s_h^{n,i}, a_h^{n,i}) - \widehat{P}_h(s_{h+1}^{n,i} | s_h^{n,i}, a_h^{n,i}) \right\|.$$

This minimax formulation effectively searches for a reward function $r$ and transition model $P_h$ within the data-consistent model class by avoiding large discrepancy from maximum likelihood models, and then performs model-based planning using the searched conservative models. In (3.1), the parameters $\lambda_1$ and $\lambda_2$ are user-specified and control the degree of conservatism encoded in the learned models. Notably, MCP remains tractable—unlike existing approaches that encode conservatism through the unmeasurable width of confidence sets via constrained optimization, which often leads to intractability.

## 3.2 Algorithm Implementation

In this section, we present the details of the implementation of the MCP algorithm, building upon the above-formulated objective function. We will establish the rigorous theoretical guarantees for Algorithm 1 later.

**Model Estimation (Lines 2–3).** The algorithm begins by estimating the reward model $\widehat{r}$ and the transition model $\widehat{P}_h$ through maximizing the log-likelihood function based on the offline dataset $\mathcal{D}$.

**Conservative Planning via Relative Performance (Line 5).** In this step, MCP enforces consistency with the offline data by regularizing the discrepancy between the learned reward and transition models and their maximum likelihood estimators. It then implicitly encodes conservatism by performing conservative evaluation—planning under the worst-case model based on the relative performance of the distributions induced by $\pi_t$ and the reference policy. This model-based evaluation avoids learning an additional value function, improving computational efficiency.

**Policy Improvement (Line 6).** MCP improves the policy without explicitly searching over a policy function class $\Pi$. Instead, it performs a mirror descent update, which is often utilized in online settings [Haarnoja et al., 2018; Geist et al., 2019], which bridges the gap between the policy space and the reward and transition model classes. As a result, the policy is no longer searched independently of $\mathcal{R}$ and $\{\mathcal{P}_h\}_{h=0}^{H-1}$.

---

**Algorithm 1** **M**odel-based **C**onservative **P**lanning (MCP)

---

**Input:** Offline data $\mathcal{D}$, regularization parameters $\lambda_1, \lambda_2$, learning rate $\eta$, reference distribution $\mu_{ref}$
1: Initialize policy $\pi_1$ as the uniform policy.
2: **Learn reward:** $\widehat{r} = \mathrm{argmax}_{r \in \mathcal{R}} \sum_{n=1}^{N} \log P_r(o = o^n | \tau^{n,1}, \tau^{n,0})$.
3: **Learn transition kernel:**
$$\widehat{P}_h = \mathrm{argmax}_{P_h \in \mathcal{P}_h} \sum_{n=1}^{N} \sum_{i=0}^{1} \log P_h(s_{h+1}^{n,i} | s_h^{n,i}, a_h^{n,i}), \ \forall h \in [0:H-1].$$
4: **for** $t = 1, 2, \ldots, T$ **do**
5:      Obtain the conservative models: $r_t, \{P_h^t\}_{h=0}^{H-1}$,
$$r_t, \{P_h^t\}_{h=0}^{H-1} \leftarrow \underset{r \in \mathcal{R}, \{P_h \in \mathcal{P}_h\}_{h=0}^{H-1}}{\arg\min} J(\pi_t; r, \{P_h\}_{h=0}^{H-1}) - \mathbb{E}_{\tau \sim \mu_{ref}}[r(\tau)]$$
$$+ \lambda_1 \mathcal{E}_1(r; \mathcal{D}) + \lambda_2 \mathcal{E}_2(\{P_h\}_{h=0}^{H-1}; \mathcal{D}).$$
6:      Update $\pi_t$ by: $\pi_{t+1}(a|s) \propto \pi_t(a|s) \exp\left( \eta \mathbb{E}_{d^{\pi_t}_{\{P_h^t\}_{h=0}^{H-1}}}[r_t(\tau) | s, a] \right)$.
7: **end for**
8: Output $\pi^{\mathrm{ALG}} := \mathrm{MixIter}(\{\pi_t\}_{t=1}^{T})$.      ▷ *mixing $\pi_1, \ldots, \pi_T$ over all iterations uniformly*

---

## 4 THEORETICAL ANALYSIS

In this section, we establish the regret guarantee for the policy returned by MCP in Algorithm 1 under the partial coverage condition and general function approximation. To start with, we introduce the notation of the concentrability coefficient to characterize the condition for the partial data coverage. For comprehensive technical details regrading this section, please refer to Appendix B.

**Definition 1** (Concentrability Coefficient for Reward). *For a comparator policy $\pi$, we define the concentrability coefficient w.r.t. the reward function class $\mathcal{R}$, and a reference policy $\mu_{ref}$:*

$$\mathfrak{C}_R(\pi) = \sup_{r \in \mathcal{R}} \frac{\mathbb{E}_{\tau^1 \sim d^\pi, \tau^0 \sim \mu_{ref}}\left[\left\| (r(\tau^1) - r(\tau^0)) - (r^\star(\tau^1) - r^\star(\tau^0)) \right\|\right]}{\mathbb{E}_{\tau^1 \sim \mu, \tau^0 \sim \mu}\left[\left\| (r(\tau^1) - r(\tau^0)) - (r^\star(\tau^1) - r^\star(\tau^0)) \right\|\right]}.$$

**Definition 2** (Concentrability Coefficient for Transition). *For a comparator policy $\pi$, we define the concentrability coefficient w.r.t. the transition function class $\{P_h\}_{h=0}^{H-1}$, and a reference policy $\mu_{ref}$:*

$$\mathfrak{C}_P(\pi) = \max_{h \in [0:H-1]} \sup_{P_h \in \mathcal{P}_h} \frac{\mathbb{E}_{(s,a) \sim d_h^\pi}\left[ D_{TV}\left( P_h(\cdot|s,a), P_h^\star(\cdot|s,a) \right) \right]}{\mathbb{E}_{(s,a) \sim \mu_h}\left[ D_{TV}\left( P_h(\cdot|s,a), P_h^\star(\cdot|s,a) \right) \right]}.$$

We should note that the finite concentrability coefficients implies the single-policy concentrability that offline data covers a single good comparator policy (e.g., the optimal policy). The single-policy concentrability is the minimum condition for the offline data coverage [Chen & Jiang, 2022; Zhan et al., 2022] in existing literature. In addition, when the reference distribution $\mu_{ref}$ is set to $\mu$, the coefficients $\mathfrak{C}_P(\pi)$ and $\mathfrak{C}_R(\pi)$ is upper bounded by the vanilla density ratio-based concentrability coefficients, i.e., $\mathfrak{C}_P(\pi) \leq \sup_{(s,a,h)} \frac{d_h^\pi(s,a)}{\mu_h(s,a)}$ and $\mathfrak{C}_R(\pi) \leq \sup_{\tau \in \mathscr{T}} \frac{d^\pi(\tau)}{\mu(\tau)}$, respectively. Before we present the main results, we impose some regular assumptions on the function classes.

**Assumption 1** (Realizability). *$r^\star \in \mathcal{R}$ and $P_h^\star \in \mathcal{P}_h, \forall h \in [0:H-1]$.*

**Assumption 2** (Boundedness). *$0 \leq r(\tau) \leq R_{\max}, \forall r \in \mathcal{R}, \tau \in \mathscr{T}$.*

**Theorem 4.1.** *Suppose Assumptions 1 and 2 hold. We set the learning rate $\eta = \sqrt{\log |\mathcal{A}|/(2R_{\max}^2 T)}$ and set the regularization coefficients $\lambda_1 = \mathcal{O}(\mathfrak{C}_R(\pi))$ and $\lambda_2 = \mathcal{O}(R_{\max}\sqrt{\mathfrak{C}_P(\pi)M_P})$. Then, for*

any comparator policy $\pi \in \Pi$ with finite $\mathfrak{C}_R(\pi)$ and $\mathfrak{C}_P(\pi)$, with probability at least $1 - \delta$, the return policy $\pi^{ALG}$ yielded by Algorithm 1 after $T$ iterations satisfies that

$$J(\pi) - J(\pi^{ALG}) \leq \mathcal{O}\left(R_{\max}\sqrt{\frac{\log|\mathcal{A}|}{T}}\right) + \mathcal{O}\left(\kappa R_{\max}\mathfrak{C}_R(\pi)\sqrt{\frac{\log(\mathcal{N}_{\mathcal{R}}(1/N)/\delta)}{N}}\right)$$
$$+ \mathcal{O}\left(R_{\max}(\mathfrak{C}_P(\pi) + M_P)H\sqrt{\frac{\log(\mathcal{N}_{\mathcal{P}}(1/N)/\delta)}{N}}\right),$$

where $M_P$ is to measure the distributional shift between $d_h^{\pi^t}$ and $\mu_h$ under the sequence of policies yielded in the mirror-descent trajectory of Algorithm 1 for $t = [1:T]$ and $h = [0:H-1]$:

$$M_P = \max_{t \in [1:T]} \max_{h \in [0:H-1]} \frac{\mathbb{E}_{(s,a) \sim d_h^{\pi^t}}[D_{TV}(P_h^t(\cdot|s,a), P_h^\star(\cdot|s,a))]}{\mathbb{E}_{(s,a) \sim \mu_h}[D_{TV}(P_h^t(\cdot|s,a), P_h^\star(\cdot|s,a))]}.$$

Theorem 4.1 implies that the policy $\pi^{ALG}$ is the "best-effort" policy in single-policy concentrability, which can compete with any good comparator policy (including the optimal policy if it is covered by offline data). In the upper bound of Theorem 4.1, the regret can be split into three parts. The last two terms correspond to the statistical errors, which are amplified by the coverage of the offline dataset. In general, the small $\mathfrak{C}_P(\pi)$ and $\mathfrak{C}_R(\pi)$ results in better statistical error guarantee. In contrast, the large ones potentially pay high variance and statistical errors. Interestingly, we find that the distributional shift measurement $M_P$ influences the regret, indicating that the mirror-descent trajectory indeed matters in Algorithm 1. Note that the first term, i.e., the optimization error $\mathcal{O}(R_{\max}\sqrt{\log|\mathcal{A}|/T})$, can be reduced with the increase of the iterations $T$. Since the optimization error is well-controlled, this regret guarantee is mainly determined by the last two statistical errors.

In many high-stakes applications, the safe policy improvement guarantees are of concern, i.e., the return policy $\pi^{ALG}$ is no worse than the behavior policy for generating offline data [Levine et al., 2020]. In the following, we show that Algorithm 1 guarantees the safe policy improvement.

**Corollary 4.2** (Safe policy improvement). *Under the conditions of Theorem 4.1, we run Algorithm 1 many iterations, i.e., $T$ is sufficiently large, with probability at least $1 - \delta$, the regret $J(\pi_b) - J(\pi^{ALG})$ is upper bounded by*

$$\mathcal{O}\left(R_{\max}\kappa\sqrt{\frac{\log(\mathcal{N}_{\mathcal{R}}(1/N)/\delta)}{N}}\right) + \mathcal{O}\left(R_{\max}HM_P\sqrt{\frac{\log(\mathcal{N}_{\mathcal{P}}(1/N)/\delta)}{N}}\right).$$

Next, we study the sample complexity of MCP and demonstrate that our algorithm achieves improved performance in comparison to prior works.

**Corollary 4.3** (Sample complexity). *Under the conditions of Theorem 4.1, the $\pi^{ALG}$ in Algorithm 1 satisfies $J(\pi) - J(\pi^{ALG}) \leq R_{\max}\varepsilon$ with probability at least $\geq 1 - \delta$, if the sample size attains*

$$\mathcal{O}\left(\frac{\kappa^2\mathfrak{C}_R^2(\pi)\log(\mathcal{N}_{\mathcal{R}}(1/N)/\delta)}{\epsilon^2} + \frac{H^2(\mathfrak{C}_P(\pi) + M_P)^2\log(\mathcal{N}_{\mathcal{P}}(1/N)/\delta)}{\epsilon^2}\right).$$

In comparison to some prior works, Kang & Oh [2025] requires sample complexity:

$$\mathcal{O}\left(\max\left\{\frac{R_{\max}^4 H^5 \log|\mathcal{A}|\log(H|\mathcal{F}|/\delta)}{\epsilon^4}, \frac{R_{\max}^2 H^2 \log(H|\mathcal{P}|/\delta)}{\epsilon^2}\right\} + \frac{\mathfrak{C}_{TR}^2\kappa^2 H \log(|\mathcal{R}|/\delta)}{\epsilon^2}\right),$$

where $|\cdot|$ denotes the cardinality of the value function class, $\mathfrak{C}_{TR} = \sup_{\tau \in \mathscr{T}} \frac{d^\pi(\tau)}{\mu(\tau)}$. This implies that the results in Kang & Oh [2025] are restricted to finite classes, but we can handle the infinite classes. Moreover, the bound in Kang & Oh [2025] is dependent on an extra model class $\mathcal{F}$, and we avoid this additional complexity via leveraging a relative-performance model-based planning procedure. Also, the MCP is more sample efficient, i.e., $\mathcal{O}(\epsilon^{-2})$ vs $\mathcal{O}(\epsilon^{-4})$. In comparison to the work Zhan et al. [2023a], they attain a sample complexity of the same order as ours, but it requires explicitly solving for the confidence set radius, which is computationally intractable. In contrast, MCP achieves both statistical efficiency and computational efficiency by avoiding constructing explicit confidence sets.

# 5 SPECIALIZED STRUCTURES ON DYNAMICS

In the previous section, the results were established for general function approximation. In the following, we consider structured dynamics and show that MCP can exploit additional structural properties to refine the model-based concentrability coefficients into more interpretable and natural quantities, leading to improved regret bounds. Specifically, we analyze two representative examples: (1) Kernelized nonlinear regulators (KNRs [Kakade et al., 2020]), which capture smooth nonlinear dynamics common in control and robotics, and (2) Factored model [Kearns & Koller, 1999], particularly effective in dealing with high-dimensional environments, e.g., via conditional independence. For comprehensive technical details regrading this section, please refer to Appendix D and E.

## 5.1 KERNELIZED NONLINEAR REGULATORS

A kernelized nonlinear regulator is a model that assumes the next state is a linear transformation of a nonlinear embedding of the current state and action, corrupted by Gaussian noise [Kakade et al., 2020]. Formally, the transition model is given by: $s' = W^*\phi(s,a) + \epsilon, \epsilon \sim \mathcal{N}(0, \zeta^2 I)$, where $\zeta \in \mathbb{R}$, $s \in \mathbb{R}^{d'}$, $a \in \mathbb{R}^{d_a}$, and $\phi : \mathcal{S} \times \mathcal{A} \to \mathbb{R}^d$ is a nonlinear feature mapping. The true model is parameterized by the unknown weight matrix $W^*$, and the corresponding class of models is indexed by $W$, so that each candidate model is denoted by $P_W$. To establish the regret guarantee for MCP in the KNRs setting, we define a new concentrability coefficient that takes advantage of the structure of KNRs. Let $\Sigma_\pi = \mathbb{E}_{(s,a)\sim d^\pi}[\phi(s,a)\phi(s,a)^\top]$, $\Sigma_\mu = \mathbb{E}_{(s,a)\sim\mu}[\phi(s,a)\phi(s,a)^\top]$ and $\Sigma_n = \sum_{i=1}^n \phi(s_i,a_i)\phi^\top(s_i,a_i) + \lambda I$. We define the relative condition number as:

$$\mathfrak{C}_P^K(\pi) = \sup_{x\in\mathbb{R}^d}\left(\frac{x^\top\Sigma_\pi x}{x^\top\Sigma_\mu x}\right).$$

We should note that $D_{TV}(P_W(\cdot|s,a), P_{W^*}(\cdot|s,a)) = c\left(\|(W-W^*)\phi(s,a)\|_2\right)$ [Devroye et al., 2018], where $c$ is a universal constant. It is easy to show that $\mathfrak{C}_P(\pi)$ is upper-bounded by the relative condition number $\mathfrak{C}_P^K(\pi)$. We now tailor Algorithm 1 to the KNR setting. We need to modify the maximum likelihood estimator of the transition model to a kernelized nonlinear regularized variant, i.e.,

$$\widehat{W} = \underset{W\in\mathbb{R}^{d'\times d}}{\arg\min}\,\mathbb{E}_{\mathcal{D}}[\|W\phi(s,a)-s'\|_2^2] + \lambda\|W\|_F^2,$$

where $\|W\|_F$ is the Frobenius norm of $W$. Now we are prepared to state the regret bound for MCP in the KNR setting.

**Theorem 5.1** (PAC Bound in KNRs). *Under the conditions of Theorem 4.1, suppose $\|\phi(s,a)\|_2 \le 1$, $\|W\|_2^2 = \mathcal{O}(1)$, $\zeta^2 = \mathcal{O}(1)$, $\lambda = \mathcal{O}(1)$, and $\|W\|_F \le c_W$ hold. We set $\lambda_1 = \mathcal{O}(\mathfrak{C}_R(\pi))$ and $\lambda_2 = \mathcal{O}(R_{\max}\sqrt{\mathfrak{C}_P(\pi)M_P^K})$. For any good comparator policy $\pi$, with probability at least $1-\delta$, the yielded $\pi^{ALG}$ by the KNR-modified Algorithm 1 satisfies that*

$$J(\pi) - J(\pi^{ALG}) \le \mathcal{O}\left(R_{\max}\sqrt{\frac{\log|\mathcal{A}|}{T}}\right) + \mathcal{O}\left(\kappa R_{\max}\mathfrak{C}_R(\pi)\sqrt{\frac{\log(\mathbb{N}_\mathcal{R}/\delta)}{N}}\right)$$

$$+ \mathcal{O}\left(R_{\max}(\mathfrak{C}_P^K(\pi)+M_P^K)H\xi\left(\lambda_{\Sigma_n^{-1}}\sqrt{\frac{\log(H\mathbb{N}_\mathcal{P}/\delta)}{N}} + \Gamma(N,\delta)\right)\right),$$

*where the coefficients $\lambda_{\Sigma_n^{-1}} = \sqrt{\lambda_{\max}(\Sigma_n^{-1})}$, $M_P^K = \max_{t\in[1:T]}\sup_{x\in\mathbb{R}^d}\left(\frac{x^\top\Sigma_{\pi_t}x}{x^\top\Sigma_\mu x}\right)$, $\Gamma(N,\delta) = \sqrt{\mathrm{rank}(\Sigma_\mu)\{\log(\exp(\mathrm{rank}(\Sigma_\mu))/\delta)\}/N}$, and $\xi = \|W^*\|_2 + d'\min(d, \mathrm{rank}(\Sigma_\mu)(\mathrm{rank}(\Sigma_\mu) + \log(1/\delta)))\log(1+N)$. The complexity of the function class in the KNR settings is characterized as $\mathbb{N}_\mathcal{P} = \mathrm{rank}(\Sigma_\mu)d'\log(1+2c_W N)$, $\mathbb{N}_\mathcal{R} = d'\log(1+2N)$.*

In comparison to Theorem 4.1, the main contribution of the modified algorithm MCP-KNR is on exploiting the KNR structures and improving the regret bound. In particular, the appearance of $\mathrm{rank}(\Sigma_\mu)$ in the bound, instead of the feature dimension $d$, ensures that the result adapts to the complexity of the data distribution. Notably, the bound holds even when $d$ is infinite, provided that the data concentrates on a low-dimensional subspace.

## 5.2 FACTORED MODELS

Factored models provide a compact representation for large-scale MDPs by exploiting structure in the state space [Osband & Van Roy, 2014]. Let $d \in \mathbb{N}^+$ and $\mathcal{B}$ be a small finite set. Rather than modeling the full transition probability over all state variables, they decompose the state $s \in \mathbb{R}^d$ into components and assume that each component $s[i]$ of the next state depends only on a small set of parent variables $\mathscr{P}_i \subseteq [1:d]$. This yields a transition function of the form $P(s'|s, a) = \prod_{i=1}^{d} P_i(s'[i]|s[\mathscr{P}_i], a)$, significantly reducing the number of parameters compared to the unfactored case. This factorization drastically reduces the model complexity: the number of parameters for the transition function becomes $L_p := \sum_{i=1}^{d} |\mathcal{A}| \cdot |\mathcal{B}|^{1+|\mathscr{P}_i|}$, as opposed to the exponential size $\mathcal{O}(\mathcal{B}^d)$ in the fully connected case, enabling more sample-efficient learning from limited data. To adapt Algorithm 1 to this structured setting and formulate a new algorithm—MCP-Factored—we modify the maximum likelihood estimation step by estimating each transition component $P_i$ separately via $\widehat{P}_i = \arg\max_P \mathbb{E}_{\mathcal{D}} \left[ \log P\left( s'[i]|s[\mathscr{P}_i], a \right) \right]$, and reconstruct the full transition model as $\widehat{P} = \prod_{i=1}^{d} \widehat{P}_i$. Before we present the regret guarantee for MCP-Factored, let us define a new concentrability coefficient. Instead of using the density ratio over the full state space, we measure it locally for each factor of the transition model. Specifically, for any comparator policy $\pi$, we define

$$\mathfrak{C}_P^F(\pi) = \max_{i \in [1:d]} \mathbb{E}_{(s,a) \sim \mu} \left[ \left( \frac{d^\pi(s[\mathscr{P}_i], a)}{\mu(s[\mathscr{P}_i], a)} \right)^2 \right].$$

We should note that this factored concentrability coefficient $\mathfrak{C}_P^F(\pi)$ relaxes the density ratio-based single-policy concentrability coefficient, i.e., $\mathfrak{C}_P^F(\pi) \leq \sup_{(s,a,h)} \frac{d_h^\pi(s,a)}{\mu_h(s,a)}$. We now present the regret bound for the MCP-Factored algorithm.

**Theorem 5.2** (PAC Bound in Factored Models). *Under the conditions of Theorem 4.1, we set* $\lambda_1 = \mathcal{O}(\mathfrak{C}_R(\pi))$, $\lambda_2 = \mathcal{O}(R_{\max}\sqrt{\mathfrak{C}_P(\pi)M_P^F})$. *For any good comparator policy* $\pi$, *with probability at least* $1 - \delta$, *after sufficiently large number of iterations, the yielded* $\pi^{ALG}$ *by the Factored-modified Algorithm 1 satisfies that the regret* $J(\pi_b) - J(\pi^{ALG})$ *is upper bounded by*

$$\mathcal{O}\left( \kappa \mathfrak{C}_R(\pi) R_{\max} \sqrt{\frac{\log(rL_r/\delta)}{N}} \right) + \mathcal{O}\left( \left( \mathfrak{C}_P^F(\pi) + M_P^F \right) HR_{\max} \sqrt{\frac{dL_p \log(L_p N d/\delta)}{N}} \right).$$

*Here,* $M_P^F = \max_{t \in [1:T]} \max_{i \in [1:d]} \mathbb{E}_{(s,a) \sim \mu} \left[ \left( \frac{d^\pi(s[\mathscr{P}_i], a)}{\mu(s[\mathscr{P}_i], a)} \right)^2 \right]$ *and* $L_r = \sum_{i=1}^{d} |\mathcal{A}| \cdot |\mathcal{B}|^{1+|\mathcal{U}_i^a|}$, *where* $\mathcal{U}_i^a$ *denotes the effective dimension of the state variables on which the reward model depends.*

The key observation in Theorem 5.2 is as follows: Instead of depending on the full state space as in the results for general function approximation, MCP-Factored leverages the structure of factored models and therefore the PAC bound scales with the number of factors and the sizes of their parent sets, summarized by the complexity term $L_p$ and $L_r$. This avoids the exponential dependence on the state dimension $d$ seen in the non-factored models.

## 6 EXPERIMENTS

**Benchmarks and Evaluation.** We evaluate the algorithmic performance of MCP on Meta-World benchmark datasets with preference feedback [Yu et al., 2020a; Kang & Oh, 2025]. We refer the readers to the Appendix F for details of benchmarks. In Table 2, we summarize the success rates of MCP and several competitive baselines, including Oracle (IQL [Kostrikov et al., 2021] trained on ground-truth explicit rewards), MR [Kim et al., 2023], PT [Kim et al., 2023], DPPO [An et al., 2023], IPL [Hejna & Sadigh, 2023], and APPO [Kang & Oh, 2025]. Most notably, MCP achieves the best mean performance for 8 tasks with varying sizes of preference samples. Another noteworthy observation is that MCP overwhelmingly outperforms APPO for the majority of the tasks. Although MCP and APPO are all model-based algorithms, APPO leverages the extra value function modeling to locally smooth over gaps in data coverage. This smoothing-based mitigation is still not sufficient to guarantee robustness in poor data coverage scenarios. Due to page limits, we provide the additional empirical results and implementation details in Appendix G.

**Effect of Dataset Size.**   To evaluate the performance of MCP under varying amounts of preference data—especially in small-data regimes—we train the model using different numbers of preferences, ranging from 100 to 2000, as shown in Figure 1(a). Notably, even with extremely limited data, i.e., $N = 100$, MCP maintains a relatively high success rate, highlighting its potential for applications with small preference datasets, such as those in medicine and finance.

Table 2: Below reports the success rates on the Meta-World medium-replay benchmark using 500 and 1000 preference-based feedback samples, averaged across three random seeds. Baseline results are from the respective papers. Note that the top two algorithms in each experiment are highlighted in bold, with the best-performing algorithm additionally shaded in light gray.

| Dataset & Methods | Oracle | MR | PT | DPPO | IPL | APPO | MCP |
|---|---|---|---|---|---|---|---|
| BPT-500 | $88.33_{\pm4.76}$ | $10.08_{\pm7.57}$ | $22.87_{\pm9.06}$ | $3.93_{\pm4.34}$ | $34.73_{\pm13.9}$ | $\mathbf{53.52}_{\pm13.9}$ | $\mathbf{56.00}_{\pm14.1}$ |
| box-close-500 | $93.40_{\pm3.10}$ | $\mathbf{29.12}_{\pm11.3}$ | $0.33_{\pm1.16}$ | $10.20_{\pm11.5}$ | $5.93_{\pm5.81}$ | $\mathbf{18.24}_{\pm15.60}$ | $7.20_{\pm3.87}$ |
| sweep-500 | $98.33_{\pm1.87}$ | $\mathbf{86.96}_{\pm6.93}$ | $43.07_{\pm24.6}$ | $10.47_{\pm15.8}$ | $27.20_{\pm23.8}$ | $26.80_{\pm5.32}$ | $11.47_{\pm9.93}$ |
| BPT-wall-500 | $56.27_{\pm6.32}$ | $0.32_{\pm0.30}$ | $0.87_{\pm1.43}$ | $0.80_{\pm1.51}$ | $8.93_{\pm9.84}$ | $\mathbf{64.32}_{\pm21.0}$ | $\mathbf{64.53}_{\pm10.6}$ |
| dial-turn-500 | $75.40_{\pm5.47}$ | $\mathbf{61.44}_{\pm6.08}$ | $68.67_{\pm12.4}$ | $26.67_{\pm22.2}$ | $31.53_{\pm12.5}$ | $\mathbf{80.96}_{\pm4.49}$ | $38.67_{\pm7.65}$ |
| sweep-into-500 | $78.80_{\pm7.96}$ | $28.40_{\pm5.47}$ | $20.53_{\pm8.26}$ | $23.07_{\pm7.02}$ | $\mathbf{32.20}_{\pm7.35}$ | $24.08_{\pm5.91}$ | $\mathbf{33.07}_{\pm4.71}$ |
| drawer-open-500 | $100.00_{\pm0.00}$ | $\mathbf{98.00}_{\pm2.32}$ | $88.73_{\pm11.6}$ | $35.93_{\pm11.2}$ | $19.00_{\pm13.6}$ | $87.68_{\pm10.0}$ | $\mathbf{98.67}_{\pm0.94}$ |
| lever-pull-500 | $98.47_{\pm1.77}$ | $79.28_{\pm2.95}$ | $\mathbf{82.40}_{\pm22.7}$ | $10.13_{\pm12.2}$ | $31.20_{\pm18.5}$ | $75.76_{\pm7.17}$ | $\mathbf{80.27}_{\pm3.10}$ |
| *Average Rank -500* | – | 3.000 | 3.500 | 5.250 | 4.000 | 2.875 | **2.375** |
| BPT-1000 | $88.33_{\pm4.76}$ | $8.48_{\pm5.80}$ | $18.27_{\pm10.6}$ | $3.20_{\pm3.04}$ | $36.67_{\pm17.4}$ | $\mathbf{59.04}_{\pm19.0}$ | $\mathbf{62.93}_{\pm17.8}$ |
| box-close-1000 | $93.40_{\pm3.10}$ | $\mathbf{27.04}_{\pm14.5}$ | $2.27_{\pm2.86}$ | $9.33_{\pm6.90}$ | $6.73_{\pm8.41}$ | $\mathbf{34.24}_{\pm18.5}$ | $10.53_{\pm5.78}$ |
| sweep-1000 | $98.33_{\pm1.87}$ | $\mathbf{87.52}_{\pm7.87}$ | $29.13_{\pm14.8}$ | $8.73_{\pm16.4}$ | $\mathbf{38.33}_{\pm24.9}$ | $17.36_{\pm12.4}$ | $20.80_{\pm11.91}$ |
| BPT-wall-1000 | $56.27_{\pm6.32}$ | $0.48_{\pm0.47}$ | $2.13_{\pm2.96}$ | $0.27_{\pm0.85}$ | $14.07_{\pm11.5}$ | $\mathbf{62.96}_{\pm18.4}$ | $\mathbf{69.73}_{\pm15.1}$ |
| dial-turn-1000 | $75.40_{\pm5.47}$ | $\mathbf{69.44}_{\pm7.00}$ | $68.80_{\pm5.50}$ | $36.40_{\pm21.9}$ | $43.93_{\pm13.4}$ | $\mathbf{81.44}_{\pm6.73}$ | $48.40_{\pm5.82}$ |
| sweep-into-1000 | $78.80_{\pm7.96}$ | $26.00_{\pm5.53}$ | $20.27_{\pm7.84}$ | $23.33_{\pm7.30}$ | $\mathbf{30.40}_{\pm7.74}$ | $18.16_{\pm11.1}$ | $\mathbf{34.13}_{\pm3.27}$ |
| drawer-open-1000 | $100.00_{\pm0.00}$ | $98.40_{\pm2.82}$ | $95.32_{\pm4.26}$ | $36.47_{\pm7.90}$ | $28.53_{\pm18.4}$ | $\mathbf{98.56}_{\pm2.68}$ | $\mathbf{99.07}_{\pm0.38}$ |
| lever-pull-1000 | $98.47_{\pm1.77}$ | $\mathbf{88.96}_{\pm3.28}$ | $72.93_{\pm10.2}$ | $8.53_{\pm9.96}$ | $40.40_{\pm17.4}$ | $76.96_{\pm4.40}$ | $\mathbf{83.33}_{\pm5.51}$ |
| *Average Rank -1000* | – | 2.750 | 4.125 | 5.375 | 3.875 | 2.750 | **2.125** |
| **Average Rank** | – | 2.875 | 3.812 | 5.312 | 3.938 | 2.813 | **2.250** |

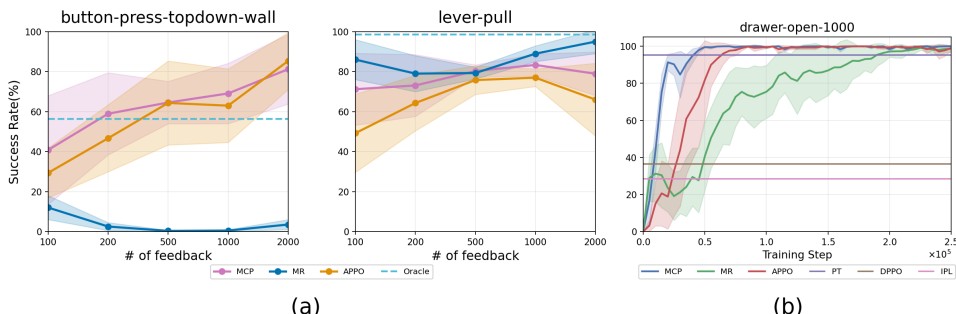

(a)                                                                                           (b)

Figure 1: (a) Model performance varying with dataset size, i.e., the number of preference feedback, ranging from 100 to 2000. (b) Training curves for drawer-open with 1000 preferences. We evaluate the results over three seeds, and the horizontal lines indicate the baseline performance summarized in Table 2.

**Training Dynamics.**   We study the training dynamics of MCP for drawer-open with 1000 preferences in Figure 1(b). We evaluate the training performance every 5000 steps. Figure 1(b) demonstrates that MCP achieves comparable performance with state-of-the-art baselines in the task. Also, it shows that MCP maintains stability in the policy improvement process, especially after sufficient training steps. .

**Ablation Study.**   We investigate the contribution of each design component of MCP on the lever-pull task with 1000 preferences in Figure 2(a). The figure compares the full MCP algorithm with several variants obtained by removing the reward regularization ($\lambda_1 = 0$), removing the transition regularization ($\lambda_2 = 0$), dropping the relative performance term ("No Relative Performance"), and replacing the model-based planner with a value-based variant ("Value-based MCP"). We evaluate the success rate every 5000 training steps and report the mean and standard deviation over three

seeds. Figure 2(a) shows that full MCP rapidly reaches a high success rate and clearly outperforms all ablated variants. In particular, the variants without reward or transition regularization make little progress, indicating that the conservative modeling terms are essential for avoiding over-optimistic and unstable policies. The "No Relative Performance" variant exhibits large variability and unstable learning, which is consistent with the lack of the safe policy improvement guarantee provided by the relative-performance objective. The "Value-based MCP" variant improves in the early stage but then degrades, suggesting that relying solely on value-based updates introduces significant bias from model errors. Overall, these results confirm that all four model components are crucial for guaranteeing the consistent policy improvement.

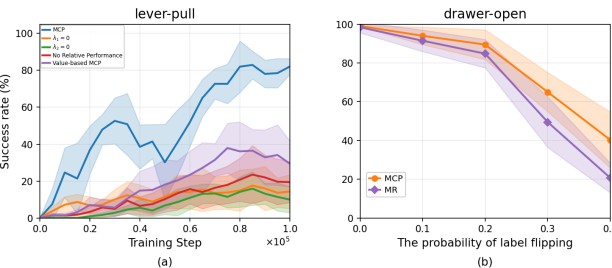

Figure 2: (a) Ablation of MCP design components on lever-pull-1000. (b) Robustness to label noise on drawer-open-1000 (performance vs. label-flipping probability).

**Robustness to Label Noise.** We further examine the robustness of MCP to noisy preference labels on the drawer-open task in Figure 2(b). Starting from a dataset with 1000 preferences, we introduce synthetic noise by independently flipping each pairwise label with probability $p$ and vary $p$ over several levels. The figure reports the resulting success rates of MCP and MR, averaged over three seeds. As the label-flipping probability increases, the performance of both methods degrades, but MCP consistently maintains higher success rates and exhibits a slower decay than MR. This result demonstrates that the conservative design of MCP provides improved robustness to label noise.

Table 3: Sensitivity of MCP to regularization hyperparameters on *drawer-open-1000* and *sweep-into-1000*.

| $\lambda_1$ | 1e-1 | 3e-1 | 1 | 3 | 1e1 |
|---|---|---|---|---|---|
| drawer-open-1000 | $94.8 \pm 3.85$ | $97.60 \pm 2.33$ | $99.07 \pm 0.38$ | $94.53 \pm 4.03$ | $86.27 \pm 10.31$ |
| sweep-into-1000 | $32.4 \pm 5.85$ | $34.13 \pm 3.27$ | $30.67 \pm 4.71$ | $24.0 \pm 9.27$ | $19.87 \pm 7.92$ |
| $\lambda_2$ | 3e-3 | 1e-2 | 3e-2 | 1e-1 | 1 |
| drawer-open-1000 | $90.93 \pm 7.30$ | $95.47 \pm 4.10$ | $99.07 \pm 0.38$ | $89.60 \pm 9.53$ | $80.13 \pm 15.19$ |
| sweep-into-1000 | $20.8 \pm 10.32$ | $23.6 \pm 5.85$ | $28.27 \pm 7.08$ | $34.13 \pm 3.27$ | $25.73 \pm 6.15$ |

**Hyperparameter sensitivity.** We conduct a hyperparameter sensitivity analysis for the regularization weights $\lambda_1, \lambda_2$ on the *drawer-open-1000* and *sweep-into-1000* tasks over three seeds. As shown in Table 3, across several orders of magnitude for each parameter, the success rate of MCP changes only mildly, indicating that our method is not highly sensitive to the exact choice of $\lambda_1, \lambda_2$.

## 7 DISCUSSION

In this work, we present the first provably sample-efficient and computationally tractable offline PbRL under partial data coverage without known dynamics. The developed algorithm implicitly encodes the conservatism into a relative performance model-based planning procedure, and guarantees the algorithmic traceability. The learned policy is able to compete with any policies within the data coverage, making it robust even in scenarios with poor data coverage. We further extend and refine the theoretical results with general function approximation under specialized dynamic structures. As a potential research direction, it is interesting to extend the Bradley–Terry–Luce framework for modeling human preferences by incorporating alternative preference models, e.g., Thurstone model.

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

# Appendix

**Table of Contents**

## A    RELATED WORK

**Offline Preference-based Reinforcement Learning.** Recent advancements in offline PbRL have investigated a variety of methods for incorporating preference feedback into policy learning. Some of these approaches rely on explicit reward modeling, while others aim to bypass it altogether. For instance, OPRL [Shin et al., 2023] introduces an active querying mechanism over offline data to infer a reward model from preferences, which is subsequently used in standard offline RL pipelines. PT [Kim et al., 2023] proposes a transformer-based reward model designed to capture non-Markovian dependencies in human feedback. In contrast, IPL [Hejna & Sadigh, 2023] avoids reward modeling entirely by directly optimizing a Q-function aligned with preferences through the inverse Bellman operator. Similarly, DPPO [An et al., 2023] formulates preference learning as a contrastive objective, enabling direct policy optimization based on preference data without estimating an intermediate reward function. While these methods demonstrate promising empirical performance, they generally lack sample complexity guarantees, which raises concerns about their theoretical robustness. To overcome these limitations, recent works have proposed offline PbRL algorithms with theoretical guarantees under partial data coverage. For instance, [Zhu et al., 2023] provides the first sample complexity results under linear reward models, while subsequent works [Zhan et al., 2023a; Pace et al., 2024] extend these to general function approximation using confidence-set-based policy optimization. Yet, constructing and optimizing over such sets is often computationally intensive. In the recent work [Kang & Oh, 2025], their algorithms either assume known transition dynamics or require fitting an extra value function that depends on the learned transition model via Bellman recursion to perform conservatism. This allows the value function to locally smooth over gaps in data coverage—if the value function is well-approximated. However, this smoothing-based mitigation strategy intrinsically requires the realizability condition on the value function and does not guarantee near-optimal regret under partial coverage. In contrast, our work proposes a model-based offline PbRL framework that encodes conservatism implicitly via relative performance objectives, avoiding confidence set construction and value function estimation. By learning both the reward and transition models from offline data and using them for conservative planning, our approach achieves PAC guarantees under general function approximation, without assuming known dynamics. Moreover, when applied to structured settings (e.g., kernelized nonlinear regulators or factored models), it yields refined generalization guarantees.

**Reinforcement Learning from Human-AI Preference.** Unlike traditional RL, which relies on explicit numerical rewards for each state-action pair, reinforcement learning from human-AI preference infers a reward function by collecting pairwise preferences over trajectories [Wirth et al., 2017; Akrour et al., 2012]. Various strategies have been proposed for eliciting preferences, typically assuming access to either a known transition model or an environment that supports interaction or rollouts [Brown et al., 2020; Christiano et al., 2017; Chen et al., 2022; Pacchiano et al., 2021; Sadigh et al., 2017; Zhan et al., 2023b; Lindner et al., 2021; Stiennon et al., 2020; Park et al., 2022; Hejna III & Sadigh, 2023]. While effective in interactive or online settings, these assumptions limit the applicability of many algorithms in the offline setting, where the agent must learn solely from a fixed dataset without further interaction.

**Offline RL.** Offline RL has gained attention because it allows learning policies without interacting with the environment, which is important in areas where safety concerns or data collection costs make interaction difficult, e.g., healthcare [Luckett et al., 2020; Li et al., 2023; Zhou et al., 2024; Jayaraman et al., 2024], autonomous driving [Yurtsever et al., 2020; Diehl et al., 2023; Lin et al., 2024], robotics [Kumar et al., 2020; Yu et al., 2021; Zhou et al., 2023; Luo et al., 2023]. However, offline RL is also difficult because the available data may not cover all relevant state-action pairs. This lack of coverage can cause errors during policy learning. To address this issue, many algorithms have been proposed that introduce conservative learning strategies to ensure reliable performance under limited data coverage [Kumar et al., 2020; Jin et al., 2021; Xie et al., 2021; Uehara & Sun, 2021b; Zhan et al., 2022; Blanchet et al., 2023; Zhou, 2023; Cetin & Celiktutan, 2023; Ding et al., 2024; Shi & Chi, 2024; Li et al., 2024]. For more discussions of offline RL, we refer the readers to [Levine et al., 2020; Prudencio et al., 2023] for detailed reviews of this topic.

# B    TECHNICAL PROOFS FOR GENERAL FUNCTION APPROXIMATION

To begin with, we will introduce some technical notations used in this section to facilitate reading.

**Summarization of Notations:**

- We use $\mathcal{R}$ to represent the function class for the reward model, $\{\mathcal{P}_h\}_{h=0}^{H-1}$ to represent the function class for transition dynamics. We use $\mathcal{N}_\mathcal{R}(1/N)$ to represent the $1/N$-bracket number for reward model function class $\mathcal{R}$, and $\mathcal{N}_\mathcal{P}(1/N)$ to represent the $1/N$-bracket number for transition dynamics function class $\{\mathcal{P}_h\}_{h=0}^{H-1}$.

- We use $r^\star$ to denote the ground-truth reward model, $\{P_h^\star\}_{h=0}^{H-1}$ to denote the ground-truth transition dynamics at time steps $h \in [0 : H-1]$. We use $\hat{r}$ and $\{\hat{P}_h\}_{h=0}^{H-1}$ to denote their corresponding MLE estimators, respectively.

- Concentrability coefficient for reward:

$$\mathfrak{C}_R(\pi) = \sup_{r \in \mathcal{R}} \frac{\mathbb{E}_{\tau^1 \sim d^\pi, \tau^0 \sim \mu_{ref}}\left[\left\|\left(r(\tau^1) - r(\tau^0)\right) - \left(r^\star(\tau^1) - r^\star(\tau^0)\right)\right\|\right]}{\mathbb{E}_{\tau^1 \sim \mu, \tau^0 \sim \mu}\left[\left\|\left(r(\tau^1) - r(\tau^0)\right) - \left(r^\star(\tau^1) - r^\star(\tau^0)\right)\right\|\right]}.$$

- Concentrability coefficient for transition:

$$\mathfrak{C}_P(\pi) = \max_{h \in [0:H-1]} \sup_{P_h \in \mathcal{P}_h} \frac{\mathbb{E}_{(s,a) \sim d_h^\pi}\left[D_{TV}\left(P_h(\cdot|s,a), P_h^\star(\cdot|s,a)\right)\right]}{\mathbb{E}_{(s,a) \sim \mu_h}\left[D_{TV}\left(P_h(\cdot|s,a), P_h^\star(\cdot|s,a)\right)\right]}.$$

- Empirical regularizer for reward model:

$$\mathcal{E}_1(r; \mathcal{D}) = \frac{1}{N} \sum_{n=1}^{N} \left\|\left(r(\tau^{n,1}) - r(\tau^{n,0})\right) - \left(\hat{r}(\tau^{n,1}) - \hat{r}(\tau^{n,0})\right)\right\|.$$

- Empirical regularizer for transition dynamics:

$$\mathcal{E}_2(\{P_h\}_{h=0}^{H-1}; \mathcal{D}) = \frac{1}{N} \sum_{n=1}^{N} \sum_{h=0}^{H-1} \sum_{i=0}^{1} \left\|P_h(s_{h+1}^{n,i}|s_h^{n,i}, a_h^{n,i}) - \hat{P}_h(s_{h+1}^{n,i}|s_h^{n,i}, a_h^{n,i})\right\|.$$

**Lemma B.1.** *Let* $M = \left(\{P_M^h\}_{h=0}^{H-1}, r_M\right)$, $M' = \left(\{P_{M'}^h\}_{h=0}^{H-1}, r_{M'}\right)$ *be two MDPs defined over the same state-action space and initial state. We define* $J_M(\pi) := J(\pi; r_M, \{P_M^h\}_{h=0}^{H-1})$. *For any policy* $\pi : \mathcal{S} \to \Delta(\mathcal{A}) \in \Pi$. *By Assumption 2, we have*

$$J_M(\pi) - J_{M'}(\pi)$$
$$\leq R_{\max} \sum_{h=0}^{H-1} \mathbb{E}_{s,a \sim d_{P_M^h}^\pi}\left[D_{TV}(P_M^h(\cdot|s,a), P_{M'}^h(\cdot|s,a))\right] + \mathbb{E}_{\tau \sim d_{\{P_M^h\}_{h=0}^{H-1}}^\pi}\left[r_M(\tau) - r_{M'}(\tau)\right].$$

***Proof of Lemma B.1.*** To analyze the difference $J_M(\pi) - J_{M'}(\pi)$, we begin by expressing it as the sum of two terms: one corresponding to the difference in rewards under the same dynamics, and another capturing the difference induced by the change in transition dynamics. Specifically, we write

$$J_M(\pi) - J_{M'}(\pi) = \mathbb{E}_{\tau \sim d^\pi_{\{P^h_M\}^{H-1}_{h=0}}}[r_M(\tau)] - \mathbb{E}_{\tau \sim d^\pi_{\{P^h_{M'}\}^{H-1}_{h=0}}}[r_{M'}(\tau)]$$

$$= \underbrace{\mathbb{E}_{\tau \sim d^\pi_{\{P^h_M\}^{H-1}_{h=0}}}[r_M(\tau) - r_{M'}(\tau)]}_{(A)}$$

$$+ \underbrace{\left(\mathbb{E}_{\tau \sim d^\pi_{\{P^h_M\}^{H-1}_{h=0}}}[r_{M'}(\tau)] - \mathbb{E}_{\tau \sim d^\pi_{\{P^h_{M'}\}^{H-1}_{h=0}}}[r_{M'}(\tau)]\right)}_{(B)}.$$

Term (A) remains unchanged. Now we aim to bound term (B). By Lemma 9 in Uehara & Sun [2021a], we have

$$|(B)| \le R_{\max} \sum^{H-1}_{h=0} \mathbb{E}_{s,a \sim d^\pi_{P^h_M}} \left[D_{TV}(P^h_M(\cdot|s,a), P^h_{M'}(\cdot|s,a)\right].$$

This completes the proof. $\square$

**Lemma B.2.** *Zhan et al. [2023a] For any reward model $r \in \mathcal{R}$, with probability at least $1 - \delta$, we have*

$$\mathbb{E}_{\tau^0 \sim \mu, \tau^1 \sim \mu}[\|(r(\tau_1) - r(\tau_0)) - (r^\star(\tau_1) - r^\star(\tau_0))\|^2] \le \frac{c\kappa^2 \log(\mathcal{N}_\mathcal{R}(1/N)/\delta)}{N},$$

*where $c > 0$ is a universal constant, $\kappa := \frac{1}{\inf_{x \in [-r_{\max}, r_{\max}]} \Phi'(x)}$ measures the nonlinearity of the link function.*

**Lemma B.3.** *Zhan et al. [2023a] With probability at least $1 - \delta$, for all $h \in [0 : H-1]$, it holds that*

$$\mathbb{E}_{(s,a) \sim \mu_h} \left[\left\|P^\star_h(\cdot|s,a) - \widehat{P}_h(\cdot|s,a)\right\|^2_{TV}\right]$$

$$\le \frac{c \log(H\mathcal{N}_\mathcal{P}(1/N)/\delta)}{N},$$

*where $c > 0$ is a universal constant.*

**Lemma B.4.** *Let $\{P^t_h\}^{H-1}_{h=0}$ be the transition models selected at line 5 of Algorithm 1 corresponding to $\pi_t$, for iterations $t \in [1 : T]$. Then with probability at least $1 - \delta$, we have*

$$\mathcal{E}_2(\{P^t_h\}^{H-1}_{h=0}; \mathcal{D}) \le \frac{cHR_{\max}M_P}{\lambda_2} \sqrt{\frac{\log(H\mathcal{N}_\mathcal{P}(1/N)/\delta)}{N}} + \mathcal{E}_2(\{P^\star_h\}^{H-1}_{h=0}; \mathcal{D}),$$

*where $c > 0$ is a universal constant.*

***Proof of Lemma B.4.*** Recall that in line 5 in Algorithm 1, the transition model $\{P^t_h\}^{H-1}_{h=0}$ is selected as minimizers of

$$J(\pi_t; r, \{P_h\}^{H-1}_{h=0}) - \mathbb{E}_{\tau \sim \mu_{ref}}[r(\tau)] + \lambda_1 \mathcal{E}_1(r; \mathcal{D}) + \lambda_2 \mathcal{E}_2(\{P_h\}^{H-1}_{h=0}; \mathcal{D}),$$

where $\pi_t$ is the policy at iteration $t$. So we have

$$J\left(\pi_t; r_t, \{P^t_h\}^{H-1}_{h=0}\right) - \mathbb{E}_{r \sim \mu_{ref}}[r_t(\tau)] + \lambda_1 \mathcal{E}_1(r_t; \mathcal{D}) + \lambda_2 \mathcal{E}_2(\{P^t_h\}^{H-1}_{h=0}; \mathcal{D})$$

$$= \min_{r \in \mathcal{R}, \{P_h \in \mathcal{P}_h\}^{H-1}_{h=0}} \left(J\left(\pi_t; r, \{P_h\}^{H-1}_{h=0}\right) - \mathbb{E}_{r \sim \mu_{ref}}[r(\tau)] + \lambda_1 \mathcal{E}_1(r; \mathcal{D}) + \lambda_2 \mathcal{E}_2(\{P_h\}^{H-1}_{h=0}; \mathcal{D})\right)$$

$$\le J\left(\pi_t; r_t, \{P^\star_h\}^{H-1}_{h=0}\right) - \mathbb{E}_{r \sim \mu_{ref}}[r_t(\tau)] + \lambda_1 \mathcal{E}_1(r_t; \mathcal{D}) + \lambda_2 \mathcal{E}_2(\{P^\star_h\}^{H-1}_{h=0}; \mathcal{D}).$$

Rearrange the equation above, and we have

$$J\left(\pi_t; r_t, \{P_h^t\}_{h=0}^{H-1}\right) - J\left(\pi_t; r_t, \{P_h^\star\}_{h=0}^{H-1}\right) \leq \lambda_2 \mathcal{E}_2(\{P_h^\star\}_{h=0}^{H-1}; \mathcal{D}) - \lambda_2 \mathcal{E}_2(\{P_h^t\}_{h=0}^{H-1}; \mathcal{D})$$

$$\leq \lambda_2 \mathcal{E}_2(\{P_h^\star\}_{h=0}^{H-1}; \mathcal{D}). \tag{B.1}$$

We also have

$$J\left(\pi_t; r_t, \{P_h^t\}_{h=0}^{H-1}\right) - \mathbb{E}_{r \sim \mu_{ref}}[r_t(\tau)] + \lambda_1 \mathcal{E}_1(r_t; \mathcal{D}) + \lambda_2 \mathcal{E}_2(\{P_h^t\}_{h=0}^{H-1}; \mathcal{D})$$

$$= \min_{r \in \mathcal{R}, P_h \in \mathcal{P}_h} \left(J\left(\pi_t; r, \{P_h\}_{h=0}^{H-1}\right) - \mathbb{E}_{r \sim \mu_{ref}}[r(\tau)] + \lambda_1 \mathcal{E}_1(r; \mathcal{D}) + \lambda_2 \mathcal{E}_2(\{P_h\}_{h=0}^{H-1}; \mathcal{D})\right)$$

$$\leq J\left(\pi_t; r_t, \{\widehat{P}_h\}_{h=0}^{H-1}\right) - \mathbb{E}_{r \sim \mu_{ref}}[r_t(\tau)] + \lambda_1 \mathcal{E}_1(r_t; \mathcal{D}) + \lambda_2 \mathcal{E}_2(\{\widehat{P}_h\}_{h=0}^{H-1}; \mathcal{D}).$$

By rearranging the equation above, we can obtain

$$\lambda_2 \mathcal{E}_2(\{P_h^t\}_{h=0}^{H-1}; \mathcal{D})$$

$$\leq \left| J\left(\pi_t; r_t, \{P_h^t\}_{h=0}^{H-1}\right) - J\left(\pi_t; r_t, \{\widehat{P}_h\}_{h=0}^{H-1}\right)\right|$$

$$= \left| J\left(\pi_t; r_t, \{P_h^t\}_{h=0}^{H-1}\right) - J\left(\pi_t; r_t, \{P_h^\star\}_{h=0}^{H-1}\right) + J\left(\pi_t; r_t, \{P_h^\star\}_{h=0}^{H-1}\right) - J\left(\pi_t; r_t, \{\widehat{P}_h\}_{h=0}^{H-1}\right)\right|$$

$$\leq \left| \lambda_2 \mathcal{E}_2(\{P_h^\star\}_{h=0}^{H-1}; \mathcal{D}) + R_{\max} \sum_{h=0}^{H-1} \mathbb{E}_{s,a \sim d_h^{\pi_t}} \left[D_{TV}\left(\widehat{P}_h(\cdot|s,a), P_h^\star(\cdot|s,a)\right)\right]\right|$$

$$\leq \lambda_2 \mathcal{E}_2(\{P_h^\star\}_{h=0}^{H-1}; \mathcal{D}) + H R_{\max} M_P \sqrt{\frac{c \log(H \mathcal{N}_P(1/N)/\delta)}{N}},$$

where $M_P = \max\limits_{t \in [1:T]} \max\limits_{h \in [0:H-1]} \frac{\mathbb{E}_{(s,a) \sim d_h^{\pi_t}}\left[D_{TV}\left(P_h^t(\cdot|s,a), P_h^\star(\cdot|s,a)\right)\right]}{\mathbb{E}_{(s,a) \sim \mu_h}\left[D_{TV}\left(P_h^t(\cdot|s,a), P_h^\star(\cdot|s,a)\right)\right]}$, the third step is by equation (B.1) and Lemma B.1. This completes the proof. $\qquad\square$

**Lemma B.5.** *Let $r^\star \in \mathcal{R}$ denote the ground -truth reward model, and let $\widehat{r}$ denote the maximum likelihood estimator. Then with probability at least $1 - \delta$, we have*

$$\mathcal{E}_1(r^\star; \mathcal{D}) = \frac{1}{N} \sum_{n=1}^{N} \left\|\left(r^\star(\tau^{n,1}) - r^\star(\tau^{n,0})\right) - \left(\widehat{r}(\tau^{n,1}) - \widehat{r}(\tau^{n,0})\right)\right\|$$

$$\leq c \sqrt{\frac{\kappa^2 \log(\mathcal{N}_\mathcal{R}(1/N)/\delta)}{N}} + c R_{\max} \sqrt{\frac{\log(\mathcal{N}_\mathcal{R}(1/N)/\delta)}{N}},$$

*where $c > 0$ is a universal constant.*

**Proof of Lemma B.5.** By Assumption 2, both $r^\star$ and $\widehat{r}$ are bounded by $R_{\max}$, so for any pair of trajectories, we have

$$\left\|\left(r^\star(\tau^{(1)}) - r^\star(\tau^{(0)})\right) - \left(\widehat{r}(\tau^{(1)}) - \widehat{r}(\tau^{(0)})\right)\right\| \leq 4 R_{\max}.$$

Define $f_r = \left(r^\star(\tau^{(1)}) - r^\star(\tau^{(0)})\right) - \left(\widehat{r}(\tau^{(1)}) - \widehat{r}(\tau^{(0)})\right).$

By Hoeffding's inequality and union bound, we obtain

$$\mathbb{P}\left(\left|\frac{1}{N} \sum_{n=1}^{N} \|f_r\| - \mathbb{E}[\|f_r\|]\right| \geq \epsilon\right) \leq 2 \mathcal{N}_\mathcal{R}(1/N) \exp\left(-\frac{2N\epsilon^2}{(4R_{\max})^2}\right).$$

Now solve for $\epsilon$ such that the RHS$\leq \delta$, we get

$$\epsilon = 4 R_{\max} \sqrt{\frac{\log(2\mathcal{N}_\mathcal{R}(1/N)/\delta)}{2N}}.$$

Meanwhile, from Lemma B.2, by Jensen's inequality, we also have

$$\mathbb{E}[\|f_r\|^2] \leq \frac{ck^2 \log(\mathcal{N}_{\mathcal{R}}(1/N)/\delta)}{N} \quad \Rightarrow \quad \mathbb{E}[\|\|f_r\|\|] \leq \sqrt{\frac{ck^2 \log(\mathcal{N}_{\mathcal{R}}(1/N)/\delta)}{N}}.$$

Putting everything together, we obtain

$$\mathcal{E}_1(r^\star; \mathcal{D}) \leq c\sqrt{\frac{\kappa^2 \log(\mathcal{N}_{\mathcal{R}}(1/N)/\delta)}{N}} + cR_{\max}\sqrt{\frac{\log(\mathcal{N}_{\mathcal{R}}(1/N)/\delta)}{N}},$$

where $c > 0$ is a universal constant. This completes the proof. $\qquad\square$

**Lemma B.6.** *Let $\{P_h^\star\}_{h=0}^{H-1}$ denote the ground-truth transition dynamics at time steps $h \in [0 : H-1]$ and let $\{\widehat{P}_h\}_{h=0}^{H-1}$ denote the maximum likelihood estimator. Then, with probability at least $1 - \delta$, we have*

$$\mathcal{E}_2(\{P_h^\star\}_{h=0}^{H-1}; \mathcal{D}) = \frac{1}{N} \sum_{n=1}^{N} \sum_{h=0}^{H-1} \sum_{i=0}^{1} \left\| P_h^\star(s_{h+1}^{n,i}|s_h^{n,i}, a_h^{n,i}) - \widehat{P}_h(s_{h+1}^{n,i}|s_h^{n,i}, a_h^{n,i}) \right\|$$

$$\leq cH\sqrt{\frac{\log(H\mathcal{N}_{\mathcal{P}}(1/N)/\delta)}{N}},$$

*where $c > 0$ is a universal constant.*

***Proof of Lemma B.6.*** Since the N trajectory pairs are sampled i.i.d, we first take expectation over the data distribution $\mu_h$, and average over $n$

$$\mathbb{E}[\mathcal{E}_2(\{P_h^\star\}_{h=0}^{H-1}; \mathcal{D})] = \sum_{h=0}^{H-1} \sum_{i=0}^{1} \mathbb{E}_{(s_h, a_h) \sim \mu_{i,h}} \left[ \left\| P_h^\star(\cdot \mid s_h, a_h) - \widehat{P}_h(\cdot \mid s_h, a_h) \right\| \right].$$

By Lemma B.3 and Jensen's inequality, summing over all $h$ and $i$ we obtain

$$\mathbb{E}[\mathcal{E}_2(\{P_h^\star\}_{h=0}^{H-1}; \mathcal{D})] \leq cH\sqrt{\frac{\log(H\mathcal{N}_{\mathcal{P}}(1/N)/\delta)}{N}},$$

where $c > 0$ is a universal constant. Applying Hoeffding's inequality and union bound over the transition dynamics function class

$$P\left( \left| \frac{1}{N} \sum_{n=1}^{N} \mathcal{E}_2(\{P_h^\star\}_{h=0}^{H-1}; \mathcal{D}) - \mathbb{E}[\mathcal{E}_2(\{P_h^\star\}_{h=0}^{H-1}; \mathcal{D})] \right| \geq \epsilon \right) \leq 2H\mathcal{N}_{\mathcal{P}}(1/N) \exp\left( -\frac{N\epsilon^2}{2H^2} \right).$$

Solve for $\epsilon$ such that the RHS is bounded by $\delta$, we obtain

$$\epsilon = H\sqrt{\frac{2\log(2H\mathcal{N}_{\mathcal{P}}(1/N)/\delta)}{N}}.$$

By applying Lemma B.4, we have

$$\mathcal{E}_2(\{P_h^t\}_{h=0}^{H-1}; \mathcal{D})$$
$$\leq \frac{cHR_{\max}M_P}{\lambda_2}\sqrt{\frac{\log(H\mathcal{N}_P(1/N)/\delta)}{N}} + cH\sqrt{\frac{\log(H\mathcal{N}_{\mathcal{P}}(1/N)/\delta)}{N}}.$$

This completes the proof. $\qquad\square$

**Lemma B.7.** *Let $\pi_t$ denote the policy at iterations $t \in [1 : T]$. For the associated reward function $r_t$, transition dynamics $\{P_h^t\}_{h=0}^{H-1}$ selected in line 5 of the Algorithm 1, we have*

$$J(\pi_t) \geq J\left(\pi_t; r_t, \{P_h^t\}_{h=0}^{H-1}\right) - \mathbb{E}_{\tau \sim \mu_{ref}}[r_t(\tau)] + \mathbb{E}_{\tau \sim \mu_{ref}}[r^\star(\tau)]$$
$$- \lambda_1 \mathcal{E}_1(r^\star; \mathcal{D}) - \lambda_2 \mathcal{E}_2(\{P_h^\star\}_{h=0}^{H-1}; \mathcal{D}).$$

***Proof of Lemma B.7.*** Please recall that $J(\pi_t) = J(\pi_t; r^\star, \{P_h^\star\}_{h=0}^{H-1})$. So we have

$$J(\pi_t) = J(\pi_t) - \mathbb{E}_{\tau \sim \mu_{ref}}[r^\star(\tau)] + \lambda_1 \mathcal{E}_1(r^\star; \mathcal{D}) + \lambda_2 \mathcal{E}_2(\{P_h^\star\}_{h=0}^{H-1}; \mathcal{D}) + \mathbb{E}_{\tau \sim \mu_{ref}}[r^\star(\tau)]$$
$$- \lambda_1 \mathcal{E}_1(r^\star; \mathcal{D}) - \lambda_2 \mathcal{E}_2(\{P_h^\star\}_{h=0}^{H-1}; \mathcal{D})$$
$$\geq \min_{r, \{P_h\}_{h=0}^{H-1}} \left(J(\pi_t; r, \{P_h\}_{h=0}^{H-1}) - \mathbb{E}_{\tau \sim \mu_{ref}}[r(\tau)] + \lambda_1 \mathcal{E}_1(r; \mathcal{D})\right.$$
$$\left. + \lambda_2 \mathcal{E}_2(\{P_h\}_{h=0}^{H-1}; \mathcal{D})\right) + \mathbb{E}_{\tau \sim \mu_{ref}}[r^\star(\tau)] - \lambda_1 \mathcal{E}_1(r^\star; \mathcal{D}) - \lambda_2 \mathcal{E}_2(\{P_h^\star\}_{h=0}^{H-1}; \mathcal{D})$$
$$= J\left(\pi_t; r_t, \{P_h^t\}_{h=0}^{H-1}\right) - \mathbb{E}_{\tau \sim \mu_{ref}}[r_t(\tau)] + \lambda_1 \mathcal{E}_1(r_t; \mathcal{D}) + \lambda_2 \mathcal{E}_2(\{P_h^t\}_{h=0}^{H-1}; \mathcal{D})$$
$$+ \mathbb{E}_{\tau \sim \mu_{ref}}[r^\star(\tau)] - \lambda_1 \mathcal{E}_1(r^\star; \mathcal{D}) - \lambda_2 \mathcal{E}_2(\{P_h^\star\}_{h=0}^{H-1}; \mathcal{D})$$
$$\geq J\left(\pi_t; r_t, \{P_h^t\}_{h=0}^{H-1}\right) - \mathbb{E}_{\tau \sim \mu_{ref}}[r_t(\tau)] + \mathbb{E}_{\tau \sim \mu_{ref}}[r^\star(\tau)]$$
$$- \lambda_1 \mathcal{E}_1(r^\star; \mathcal{D}) - \lambda_2 \mathcal{E}_2(\{P_h^\star\}_{h=0}^{H-1}; \mathcal{D}),$$

where the third step is by the optimality of the $r_t$ and $\{P_h^t\}_{h=0}^{H-1}$. This completes the proof. $\qquad\square$

Now we want to make some modifications to the Lemma B.1 to serve for proof of the main theorem later.

**Lemma B.8.** *Let $\pi \in \Pi$ be an arbitrary policy. Then, for the reward model $r_t$ selected in Line 5 of Algorithm 1 corresponding to $\pi_t$, with probability at least $1 - \delta$, for all $t \in [1 : T]$, we have*

$$\mathbb{E}_{\tau^1 \sim d^\pi, \tau^0 \sim \mu_{ref}}[\|r_t(\tau^1) - r_t(\tau^0) - (r^\star(\tau^1) - r^\star(\tau^0))\|] \leq c\mathfrak{C}_R(\pi)\sqrt{\frac{\kappa^2 \log(\mathcal{N}_\mathcal{R}(1/N)/\delta)}{N}},$$

*where $c > 0$ is a universal constant.*

***Proof of Lemma B.8.***
$$\mathbb{E}_{\tau^1 \sim d^\pi, \tau^0 \sim \mu_{ref}}[\|r_t(\tau^1) - r_t(\tau^0) - (r^\star(\tau^1) - r^\star(\tau^0))\|]$$
$$\leq \mathfrak{C}_R(\pi)\mathbb{E}_{\tau^1 \sim \mu, \tau^0 \sim \mu}[\|r_t(\tau^1) - r_t(\tau^0) - (r^\star(\tau^1) - r^\star(\tau^0))\|]$$
$$\leq c\mathfrak{C}_R(\pi)\sqrt{\frac{\kappa^2 \log(\mathcal{N}_\mathcal{R}(1/N)/\delta)}{N}},$$

where the second step is by Lemma B.2. This completes the proof. $\qquad\square$

**Lemma B.9.** *Let $\pi \in \Pi$ be an arbitrary policy. Then, for the transition model $\{P_h^t\}_{h=0}^{H-1}$ selected in Line 5 of Algorithm 1 corresponding to $\pi_t$, with probability at least $1 - \delta$, for all $t \in [1 : T]$, we have*

$$\sum_{h=0}^{H-1} \mathbb{E}_{d_h^\pi}\left[D_{TV}(P_h^t(\cdot|s, a), P_h^\star(\cdot|s, a))\right]$$
$$\leq c\mathfrak{C}_P(\pi)H\left(\frac{R_{\max}M_P}{\lambda_2}\sqrt{\frac{\log(HN_P(1/N)/\delta)}{N}} + \sqrt{\frac{\log(HN_\mathcal{P}(1/N)/\delta)}{N}}\right),$$

*where $c > 0$ is a universal constant.*

*Proof of Lemma B.9.*

$$\sum_{h=0}^{H-1} \mathbb{E}_{d_h^\pi} \left[ D_{TV}(P_h^t(\cdot|s,a), P_h^\star(\cdot|s,a)) \right]$$

$$\leq \mathfrak{C}_P(\pi) \sum_{h=0}^{H-1} \mathbb{E}_{\mu_h} \left[ D_{TV}(P_h^t(\cdot|s,a), P_h^\star(\cdot|s,a)) \right]$$

$$\leq \mathfrak{C}_P(\pi) \sum_{h=0}^{H-1} \left( \underbrace{\mathbb{E}_{\mu_h} \left[ \left\| P_h^t(\cdot|s,a), \widehat{P}_h(\cdot|s,a) \right\|_{TV} \right]}_{(a)} + \underbrace{\mathbb{E}_{\mu_h} \left[ \left\| P_h^\star(\cdot|s,a), \widehat{P}_h(\cdot|s,a) \right\|_{TV} \right]}_{(b)} \right).$$

The bound for term (b) can be obtained from Lemma B.3. Now we want to bound term (a) by Hoeffding's inequality and apply the union bound

$$\mathbb{P}\left( \left| \frac{1}{N} \sum_{n=1}^{N} \mathcal{E}_2(\{P_h^t\}_{h=0}^{H-1}; \mathcal{D}) - \sum_{h=0}^{H-1} \mathbb{E}_{\mu_h} \left[ \left\| P_h^t(\cdot|s,a), \widehat{P}_h(\cdot|s,a) \right\|_{TV} \right] \right| \geq \epsilon \right)$$

$$\leq 2H\mathcal{N}_{\mathcal{P}}(1/N) \exp\left( -\frac{N\epsilon^2}{2H^2} \right),$$

where the bound for $\mathcal{E}_2(\{P_h^t\}_{h=0}^{H-1})$ can be obtained from Lemma B.6. Solving for $\epsilon$ such that the RHS is at most $\delta$

$$\epsilon = H\sqrt{\frac{2\log(2H\mathcal{N}_{\mathcal{P}}(1/N)/\delta)}{N}}.$$

Combining all the equations above completes the proof. $\qquad\square$

We are now ready to establish the proof of the main theorem. Before we start, we define $J_{\mathcal{M}_t}(\pi) := J(\pi, r_t, \{P_h^t\}_{h=0}^{H-1})$, where $r_t$ and $\{P_h^t\}_{h=0}^{H-1}$ are conservatively estimated models corresponding to $\pi_t$ and please recall that $J(\pi) = J(\pi; r^\star, \{P_h^\star\}_{h=0}^{H-1})$.

*Proof of Theorem 4.1.*

$$J(\pi) - J(\pi^{\text{ALG}})$$

$$= \frac{1}{T} \sum_{t=1}^{T} (J(\pi) - J(\pi_t))$$

$$\leq \frac{1}{T} \sum_{t=1}^{T} \left( J(\pi) - \left( \mathbb{E}_{\tau \sim \mu_{ref}}[r^\star(\tau)] - \mathbb{E}_{\tau \sim \mu_{ref}}[r_t(\tau)] \right) - J_{\mathcal{M}_t}(\pi_t) \right)$$

$$\quad + \lambda_1 \mathcal{E}_1(r^\star; \mathcal{D}) + \lambda_2 \mathcal{E}_2(\{P_h^\star\}_{h=0}^{H-1}; \mathcal{D})$$

$$\leq \frac{1}{T} \sum_{t=1}^{T} (J_{\mathcal{M}_t}(\pi) + R_{\max} \sum_{h=0}^{H-1} \mathbb{E}_{(s,a) \sim d_h^\pi}[D_{TV}(P_h^t(\cdot|s,a), P_h^\star(\cdot|s,a))]$$

$$\quad + \mathbb{E}_{\tau \sim d^\pi}[r^\star(\tau) - r_t(\tau)] - \left( \mathbb{E}_{\tau \sim \mu_{ref}}[r^\star(\tau)] - \mathbb{E}_{\tau \sim \mu_{ref}}[r_t(\tau)] \right) - J_{\mathcal{M}_t}(\pi_t))$$

$$\quad + \lambda_1 \mathcal{E}_1(r^\star; \mathcal{D}) + \lambda_2 \mathcal{E}_2(\{P_h^\star\}_{h=0}^{H-1}; \mathcal{D})$$

$$\leq \frac{1}{T} \sum_{t=1}^{T} (J_{\mathcal{M}_t}(\pi) + R_{\max} \sum_{h=0}^{H-1} \mathbb{E}_{(s,a) \sim d_h^\pi}[D_{TV}(P_h^t(\cdot|s,a), P_h^\star(\cdot|s,a))]$$

$$\quad + \mathbb{E}_{\tau^1 \sim d^\pi, \tau^0 \sim \mu_{ref}}[\|r^\star(\tau^1) - r_t(\tau^1) - (r^\star(\tau^0) - r_t(\tau^0))\|] - J_{\mathcal{M}_t}(\pi_t))$$

$$\quad + \lambda_1 \mathcal{E}_1(r^\star; \mathcal{D}) + \lambda_2 \mathcal{E}_2(\{P_h^\star\}_{h=0}^{H-1}; \mathcal{D})$$

$$\lesssim \frac{1}{T} \sum_{t=1}^{T} (J_{\mathcal{M}_t}(\pi) - J_{\mathcal{M}_t}(\pi_t)) + \lambda_1 \mathcal{E}_1(r^\star; \mathcal{D}) + \lambda_2 \mathcal{E}_2(\{P_h^\star\}_{h=0}^{H-1}; \mathcal{D}))$$

$$+ HR_{\max}\mathfrak{C}_P(\pi)\left(\frac{R_{\max}M_P}{\lambda_2}\sqrt{\frac{\log(HN_P(1/N)/\delta)}{N}} + \sqrt{\frac{\log(HN_{\mathcal{P}}(1/N)/\delta)}{N}}\right)$$

$$+ \mathfrak{C}_R(\pi)\sqrt{\frac{\kappa^2\log(\mathcal{N}_{\mathcal{R}}(1/N)/\delta)}{N}}$$

$$\lesssim R_{\max}\sqrt{\frac{\log|\mathcal{A}|}{T}} + \lambda_1\mathcal{E}_1(r^\star;\mathcal{D}) + \lambda_2\mathcal{E}_2(\{P_h^\star\}_{h=0}^{H-1};\mathcal{D})$$

$$+ HR_{\max}\mathfrak{C}_P(\pi)\left(\frac{R_{\max}M_P}{\lambda_2}\sqrt{\frac{\log(HN_P(1/N)/\delta)}{N}} + \sqrt{\frac{\log(2HN_{\mathcal{P}}(1/N)/\delta)}{N}}\right)$$

$$+ \mathfrak{C}_R(\pi)\sqrt{\frac{\kappa^2\log(\mathcal{N}_{\mathcal{R}}(1/N)/\delta)}{N}}$$

$$\lesssim R_{\max}\sqrt{\frac{\log|\mathcal{A}|}{T}} + \lambda_1\left(\sqrt{\frac{\kappa^2\log(\mathcal{N}_{\mathcal{R}}(1/N)/\delta)}{N}} + 4R_{\max}\sqrt{\frac{\log(2\mathcal{N}_{\mathcal{R}}(1/N)/\delta)}{2N}}\right)$$

$$+ \lambda_2\left(H\sqrt{\frac{\log(HN_{\mathcal{P}}(1/N)/\delta)}{N}} + H\sqrt{\frac{2\log(2HN_{\mathcal{P}}(1/N)/\delta)}{N}}\right)$$

$$+ HR_{\max}\mathfrak{C}_P(\pi)\left(\frac{R_{\max}M_P}{\lambda_2}\sqrt{\frac{\log(HN_P(1/N)/\delta)}{N}} + \sqrt{\frac{\log(2HN_{\mathcal{P}}(1/N)/\delta)}{N}}\right)$$

$$+ \mathfrak{C}_R(\pi)\sqrt{\frac{\kappa^2\log(\mathcal{N}_{\mathcal{R}}(1/N)/\delta)}{N}},$$

where the second step is by Lemma B.7, the third step is by Lemma B.1, the fifth step is by Lemma B.9, Lemma B.8. the sixth step is by Lemma C.4, the seventh step is by Lemma B.5, Lemma B.6. Substituting $\lambda_1 = \mathcal{O}\left(\mathfrak{C}_R(\pi)\right)$, $\lambda_2 = \mathcal{O}\left(R_{\max}\sqrt{\mathfrak{C}_P(\pi)M_P}\right)$ completes the proof. $\qquad\square$

## C   OPTIMIZATION ERROR

For simplicity, we assume that the transition model $P_h^t$ is homogeneous across all steps $h \in [0 : H-1]$. That is, we write $P_t := P_h^t$ for all h. This simplification is made only for notational clarity and does not affect the correctness of the result.

For each iteration $t \in [1 : T]$, the algorithm proceeds as follows:

**Step 1. Model Selection:**

Let the reward model $\widehat{r}_t$ and transition models $P_t$ be selected by solving the following penalized objective

$$r_t, P_t = \min_{r,P} J\left(\pi_t; r, P\right) - \mathbb{E}_{\tau\sim\mu_{ref}}[r(\tau)] + \lambda_1\mathcal{E}_1(r;\mathcal{D}) + \lambda_2\mathcal{E}_2(P;\mathcal{D}).$$

**Step 2. Policy Improvement:**

Update the policy using an exponentiated update rule based on the estimated reward signal

$$\pi_{t+1}(a|s) \propto \pi_t(a|s)\exp\left(\eta\mathbb{E}_{d_{P_t}^{\pi_t}}\left[r_t(\tau)|s,a\right]\right).$$

Let $J_{\mathcal{M}_t}(\pi)$ denote the expected return under reward model $r_t$ and transition dynamics $P_t$. We define the cumulative regret over $T$ iterations as

$$\mathfrak{R}_T := \max_{\pi\in\Pi}\sum_{t=1}^{T}\left(J_{\mathcal{M}_t}(\pi) - J_{\mathcal{M}_t}(\pi_t)\right).$$

Additionally, define the entropy-like regularization function

$$\psi_s(\pi) := \frac{1}{\eta}\sum_{a\in\mathcal{A}}\pi(a|s)\log\pi(a|s).$$

**Lemma C.1.** *Let $\pi : \mathcal{S} \to \Delta(\mathcal{A}) \in \Pi$ be an arbitrary policy. Then for any state $s \in \mathcal{S}$, the following inequality holds*

$$\sum_{t=1}^{T} \left\langle \pi_{t+1}(\cdot|s), \mathbb{E}_{d_{P_t}^{\pi_t}} [\widehat{r}_t(\tau)|s, \cdot] \right\rangle - \psi_s(\pi_1) \geq \sum_{t=1}^{T} \left\langle \pi(\cdot|s), \mathbb{E}_{d_{P_t}^{\pi_t}} [\widehat{r}_t(\tau)|s, \cdot] \right\rangle - \psi_s(\pi).$$

***Proof of Lemma C.1.*** We prove the results via mathematical induction over $T$ by following [Xie et al., 2021]. When $T = 0$, both sides of the inequality are zero. This holds because no iterations are executed, and we define $\psi_s(\pi_1)$ to be the entropy of the initial uniform policy. Thus, the inequality trivially holds.

Assume the statement holds for $T = T'$. That is,

$$\sum_{t=1}^{T'} \left\langle \pi_{t+1}(\cdot|s), \mathbb{E}_{d_{P_t}^{\pi_t}} [\widehat{r}_t(\tau)|s, \cdot] \right\rangle - \psi_s(\pi_1) \geq \sum_{t=1}^{T'} \left\langle \pi(\cdot|s), \mathbb{E}_{d_{P_t}^{\pi_t}} [\widehat{r}_t(\tau)|s, \cdot] \right\rangle - \psi_s(\pi).$$

We want to prove it for $T = T' + 1$. Consider: $\sum_{t=1}^{T'+1} \left\langle \pi_{t+1}(\cdot|s), \mathbb{E}_{d_{P_t}^{\pi_t}} [\widehat{r}_t(\tau)|s, \cdot] \right\rangle - \psi_s(\pi_1)$.

We can decompose this as

$$\sum_{t=1}^{T'+1} \left\langle \pi_{t+1}(\cdot|s), \mathbb{E}_{d_{P_t}^{\pi_t}} [\widehat{r}_t(\tau)|s, \cdot] \right\rangle - \psi_s(\pi_1)$$

$$= \sum_{t=1}^{T'} \left\langle \pi_{t+1}(\cdot|s), \mathbb{E}_{d_{P_t}^{\pi_t}} [\widehat{r}_t(\tau)|s, \cdot] \right\rangle - \psi_s(\pi_1) + \left\langle \pi_{T'+2}(\cdot|s), \mathbb{E}_{d_{P_{t+1}}^{\pi_{t+1}}} [\widehat{r}_{T'+1}(\tau)|s, \cdot] \right\rangle$$

$$\geq \sum_{t=1}^{T'} \langle \pi, \mathbb{E}_{d_{P_t}^{\pi_t}} [\widehat{r}_t(\tau)|s, \cdot] \rangle - \psi_s(\pi_{T'+2}) + \left\langle \pi_{T'+2}(\cdot|s, \cdot), \mathbb{E}_{d_{P_{t+1}}^{\pi_{t+1}}} [\widehat{r}_{T'+1}(\tau)|s, \cdot] \right\rangle$$

$$= \sum_{t=1}^{T'+1} \langle \pi, \mathbb{E}_{d_{P_t}^{\pi_t}} [\widehat{r}_t(\tau)|s, \cdot] \rangle - \psi_s(\pi_{T'+2})$$

$$\geq \sum_{t=1}^{T'+1} \langle \pi, \mathbb{E}_{d_{P_t}^{\pi_t}} [\widehat{r}_t(\tau)|s, \cdot] \rangle - \psi_s(\pi),$$

where the second step is by the induction hypothesis, the fourth step is by the optimality of $\pi_{T'+2}(\cdot|s)$, i.e, $\pi_{T'+2} = \arg\max_{\pi'} \left\{ \left\langle \pi'(\cdot|s), \mathbb{E}_{d_{P_{t+1}}^{\pi_{t+1}}} [\widehat{r}_{T'+1}(\tau)|s, \cdot] \right\rangle - \psi_s(\pi') \right\}$. The proof is completed. □

**Lemma C.2.** *Let $\pi \in \Pi$ be an arbitrary policy. For any state $s \in \mathcal{S}$, The following inequality holds*

$$\sum_{t=1}^{T} \langle \pi(\cdot|s) - \pi_t(\cdot|s), \mathbb{E}_{d_{P_t}^{\pi_t}} [r_t(\tau)|s, \cdot] \rangle \leq \sum_{t=1}^{T} \langle \pi_{t+1}(\cdot|s) - \pi_t(\cdot|s), \mathbb{E}_{d_{P_t}^{\pi_t}} [r_t(\tau)|s, \cdot] \rangle - \psi_s(\pi_1).$$

***Proof of Lemma C.2.*** We begin by writing the LHS as

$$\sum_{t=1}^{T} \left\langle \pi(\cdot|s) - \pi_t(\cdot|s), \mathbb{E}_{d_{P_t}^{\pi_t}} [r_t(\tau)|s, \cdot] \right\rangle$$

$$= \sum_{t=1}^{T} \left( \left\langle \pi(\cdot|s), \mathbb{E}_{d_{P_t}^{\pi_t}} [r_t(\tau)|s, \cdot] \right\rangle - \left\langle \pi_t(\cdot|s), \mathbb{E}_{d_{P_t}^{\pi_t}} [r_t(\tau)|s, \cdot] \right\rangle \right)$$

$$= \sum_{t=1}^{T} \left\langle \pi_{t+1}(\cdot|s) - \pi_t(\cdot|s), \mathbb{E}_{d_{P_t}^{\pi_t}}\left[r_t(\tau)|s,\cdot\right]\right\rangle + \sum_{t=1}^{T} \left\langle \pi(\cdot|s) - \pi_{t+1}(\cdot|s,\cdot), \mathbb{E}_{d_{P_t}^{\pi_t}}\left[r_t(\tau)|s,\cdot\right]\right\rangle$$

$$\leq \sum_{t=1}^{T} \left\langle \pi_{t+1}(\cdot|s) - \pi_t(\cdot|s,\cdot), \mathbb{E}_{d_{P_t}^{\pi_t}}\left[\widehat{r}_t(\tau)|s,\cdot\right]\right\rangle - \psi_s(\pi_1),$$

where the last inequality is by Lemma C.1. This completes the proof. $\qquad\square$

**Lemma C.3.** *Let $\pi \in \Pi$ be an arbitrary policy. Then for any state $s \in \mathcal{S}$, if we set the learning rate $\eta = \sqrt{\frac{\log |\mathcal{A}|}{2R_{\max}^2 T}}$*

$$\sum_{t=1}^{T} \langle \pi(\cdot|s) - \pi_t(\cdot|s), \mathbb{E}_{d_{P_t}^{\pi_t}}\left[r_t(\tau)|s,\cdot\right]\rangle \leq 2R_{\max}\sqrt{2\log|\mathcal{A}|T}.$$

***Proof of Lemma C.3.*** We define the surrogate objective accumulated over $t$ iterations as

$$\mathcal{F}_{s,t}(\pi) := \sum_{t=1}^{T} \langle \pi(\cdot|s), \mathbb{E}_{P_t}\left[r_t(\tau)|s,\cdot\right]\rangle - \psi_s(\pi).$$

Let $B_{\mathcal{F}_{s,t}}(\cdot\|\cdot)$ denote the Bergman divergence with respect to $\mathcal{F}_{s,t}$. By the definition of Bergman divergence, we have

$$\begin{aligned}
\mathcal{F}_{s,t}(\pi_t) &= \mathcal{F}_{s,t}(\pi_{t+1}) + \left\langle \pi_t(\cdot|s) - \pi_{t+1}(\cdot|s), \nabla\mathcal{F}_{s,t}(\pi)|_{\pi=\pi_{t+1}}\right\rangle + B_{\mathcal{F}_{s,t}}(\pi_t\|\pi_{t+1}) \\
&\leq \mathcal{F}_{s,t}(\pi_{t+1}) + B_{\mathcal{F}_{s,t}}(\pi_t\|\pi_{t+1}) \\
&= \mathcal{F}_{s,t}(\pi_{t+1}) - B_{\psi_s}(\pi_t\|\pi_{t+1}) \\
\Rightarrow B_{\psi_s}(\pi_t\|\pi_{t+1}) &\leq \mathcal{F}_{s,t}(\pi_{t+1}) - \mathcal{F}_{s,t}(\pi_t) \\
&\leq \left\langle \pi_{t+1}(\cdot|s) - \pi_t(\cdot|s), \mathbb{E}_{d_{P_t}^{\pi_t}}\left[r_t(\tau)|s,\cdot\right]\right\rangle,
\end{aligned}$$

where the second step is because $\pi_t$ is the maximizer of $\mathcal{F}_{s,t}$, and the gradient term is non-positive, the third step is because both $\mathcal{F}_{s,t}$ and $\psi_s(\pi)$ are linear and convex, we have: $B_{\mathcal{F}_{s,t}}(\pi_t\|\pi_{t+1}) = -B_{\psi_s}(\pi_t\|\pi_{t+1})$.

To convert the Bergman divergence into a squared norm, we follow [Xie et al., 2021] by applying the second-order Taylor expansion

$$B_{\psi_s}(\pi_t\|\pi_{t+1}) = \frac{1}{2}\left\|\pi_t(\cdot|s) - \pi_{t+1}(\cdot|s)\right\|_{H_{\psi_s}(\pi_t')}^2,$$

where $\pi_t' := \alpha\pi_t + (1-\alpha)\pi_{t+1}$ for some $\alpha \in [0,1]$, and $H_{\psi_s}$ is the hessian of $\psi_s$.

Using Cauchy–Schwarz inequality

$$\left\langle \pi_{t+1} - \pi_t, \mathbb{E}_{d_{P_t}^{\pi_t}}\left[r_t(\tau)|s\right]\right\rangle \leq \|\pi_{t+1} - \pi_t\|_{H_{\psi_s}(\pi_t')} \cdot \left\|\mathbb{E}_{d_{P_t}^{\pi_t}}\left[r_t(\tau)|s,\cdot\right]\right\|_{H_{\psi_s}^{-1}(\pi_t')}.$$

From the Hessian of $\psi_s$, it is known that

$$\left\|\mathbb{E}_{d_{P_t}^{\pi_t}}\left[r_t(\tau)|s\right]\right\|_{H_{\psi_s}^{-1}} \leq \sqrt{\eta}\left\|\mathbb{E}_{d_{P_t}^{\pi_t}}\left[r_t(\tau)|s\right]\right\|_{\infty} \leq \sqrt{\eta}R_{\max}.$$

Combining everything

$$\left\langle \pi_{t+1} - \pi_t, \mathbb{E}_{d_{P_t}^{\pi_t}}\left[r_t(\tau)|s,\cdot\right]\right\rangle \leq \sqrt{2B_{\psi_s}(\pi_t\|\pi_{t+1})} \cdot \sqrt{\eta}R_{\max}$$

$$\Rightarrow \left\langle \pi_{t+1} - \pi_t, \mathbb{E}_{d_{P_t}^{\pi_t}}\left[r_t(\tau)|s,\cdot\right]\right\rangle \leq \sqrt{2}\left\langle \pi_{t+1} - \pi_t, \mathbb{E}_{d_{P_t}^{\pi_t}}\left[r_t(\tau)|s,\cdot a\right]\right\rangle \cdot \sqrt{\eta}R_{\max}$$

$$\Rightarrow \left\langle \pi_{t+1} - \pi_t, \mathbb{E}_{d_{P_t}^{\pi_t}}[r_t(\tau)|s,\cdot] \right\rangle \le 2\eta R_{\max}^2$$

$$\sum_{t=1}^{T} \left\langle \pi_{t+1} - \pi_t, \mathbb{E}_{d_{P_t}^{\pi_t}}[r_t(\tau)|s,\cdot] \right\rangle \le 2\eta R_{\max}^2 T.$$

By Lemma C.2, we have

$$\sum_{t=1}^{T} \langle \pi - \pi_t, \mathbb{E}_{d_{P_t}^{\pi_t}}[r_t(\tau)|s,\cdot] \rangle \le \sum_{t=1}^{T} \langle \pi_{t+1} - \pi_t, \mathbb{E}_{d_{P_t}^{\pi_t}}[r_t(\tau)|s,\cdot] \rangle - \psi_s(\pi_1)$$

$$\le 2\eta R_{\max}^2 T + \frac{\log |\mathcal{A}|}{\eta},$$

where the second step is because $\pi_1$ is the uniform policy. Choosing $\eta = \sqrt{\frac{\log |\mathcal{A}|}{2R_{\max}^2 T}}$ concludes the proof. $\qquad\square$

**Lemma C.4.** *Let* $\pi^{ALG} = \operatorname{argmax}_{\pi:\mathcal{S}\to\Delta(\mathcal{A})} \sum_{t=1}^{T} J_{\mathcal{M}_t}(\pi) - J_{\mathcal{M}_t}(\pi_t)$ *and* $\eta = \sqrt{\frac{\log |\mathcal{A}|}{2R_{\max}^2 T}}$, *we have*

$$\mathfrak{R}_T \le 2R_{\max}\sqrt{2T \log |\mathcal{A}|}.$$

***Proof of Lemma C.4.*** Please recall that $J_{\mathcal{M}_t}(\pi) = J(\pi, r_t, P_t)$. We apply the standard performance difference Lemma Kakade & Langford [2002], which gives

$$\mathfrak{R}_T = \sum_{t=1}^{T} J_{\mathcal{M}_t}(\pi^{\mathrm{ALG}}) - J_{\mathcal{M}_t}(\pi_t) = \sum_{t=1}^{T} \mathbb{E}_{d_{P_t}^{\pi^{\mathrm{ALG}}}}\left[\mathbb{E}_{d_{P_t}^{\pi^{\mathrm{ALG}}}}[r_t(\tau)|s,\cdot] - \mathbb{E}_{d_{P_t}^{\pi^{\mathrm{ALG}}}}[r_t(\tau)|s,\cdot]\right]$$

$$= \sum_{t=1}^{T} \mathbb{E}_{d_{P_t}^{\pi^{\mathrm{ALG}}}}\left[\langle \pi^{\mathrm{ALG}}(\cdot|s) - \pi_t(\cdot|s), \mathbb{E}_{d_{P_t}^{\pi^{\mathrm{ALG}}}}[r_t(\tau)|s,\cdot]\rangle\right]$$

$$\le 2R_{\max}\sqrt{2T \log |\mathcal{A}|},$$

where the last step is by Lemma C.3. This completes the proof. $\qquad\square$

## D  TECHNICAL PROOFS FOR KNRS

In the Kernelized Nonlinear Regulator (KNR) setting, the structural constraint is imposed on the transition probability model, rather than the reward model. Consequently, we focus on modifying the related Lemmas on the transition model accordingly. Before we begin, let's recall some definitions and notations for better readability.

**Summarization of Notations:**

- We use $d$ to represents the dimension of the feature mapping $\phi(s,a)$, while $d'$ denotes the dimension of the state space $\mathcal{S}$.
- $\Sigma_n = \sum_{i=1}^{n} \phi(s_i, a_i)\phi^\top(s_i, a_i) + \lambda I$, $\Sigma_\pi = \mathbb{E}_{(s,a)\sim d^\pi}[\phi(s,a)\phi(s,a)^\top]$, and $\Sigma_\mu = \mathbb{E}_{(s,a)\sim\mu}[\phi(s,a)\phi(s,a)^\top]$.
- Relative condition number: $\mathfrak{C}_P^K(\pi) = \sup_{x\in\mathbb{R}^d}\left(\frac{x^\top \Sigma_\pi x}{x^\top \Sigma_\mu x}\right)$.

**Lemma D.1.** *Let* $\phi : \mathcal{S} \times \mathcal{A} \to \mathbb{R}^d$ *be a feature mapping with* $\|\phi(s,a)\|_2 \le 1$, *and define the KNR transition model class*

$$\mathcal{F}_{\mathrm{KNR}} := \{f_W(s,a) = W\phi(s,a) : \|W\|_F \le L\}, \quad W \in \mathbb{R}^{d'\times d},$$

*where* $d'$ *is the dimension of the state space.*

*The following cover number bounds hold*

$$\log \mathcal{N}_\infty(\{\mathcal{P}_h\}_{h=0}^{H-1}, 1/N) \leq \operatorname{rank}(\Sigma_\mu) d' \log(1 + 2LN) := \mathbb{N}_\mathcal{P}.$$
$$\log \mathcal{N}_\infty(\mathcal{R}, 1/N) \leq \operatorname{rank}(\Sigma_\mu) \log(1 + 2N) := \mathbb{N}_\mathcal{R}.$$

***Proof of Lemma D.1.*** We begin with the observation that for any $W \in \mathbb{R}^{d' \times d}$, the function $f_W(s, a) := W\phi(s, a)$ is linear in $\phi(s, a)$.

Let $\mathcal{H}_\mu := \operatorname{Im}(\Sigma_\mu) \subseteq \mathbb{R}^d$ be the image of $\Sigma_\mu$, and let $r = \dim(\mathcal{H}_\mu) = \operatorname{rank}(\Sigma_\mu)$. By the definition of image, we have

$$\mathcal{H}_\mu = \operatorname{Span}\{\phi(s, a) : (s, a) \in \Delta(\mu)\}.$$

Hence, every $\phi(s, a)$ lies in $\mathcal{H}_\mu$, and there exists an orthonormal matrix $U \in \mathbb{R}^{d \times r}$ whose columns form a basis for $\mathcal{H}_\mu$, such that

$$\phi(s, a) = Uz(s, a), \quad z(s, a) \in \mathbb{R}^r.$$

Substituting into the function expression, we obtain

$$f_W(s, a) = W\phi(s, a) = WUz(s, a).$$

Define $\tilde{W} := WU \in \mathbb{R}^{d' \times r}$. Then we can rewrite

$$f_W(s, a) = \tilde{W}z(s, a).$$

Because $U$ has orthonormal columns, we have

$$\|\tilde{W}\|_F = \|WU\|_F \leq \|W\|_F \cdot \|U\|_2 = \|W\|_F,$$

where $\|U\|_2 = 1$. Therefore, if $\|W\|_F \leq L$, then $\|\tilde{W}\|_F \leq L$.

Thus, the function class $\mathcal{F}_{\text{KNR}}$ is equivalent to the reduced class

$$\tilde{\mathcal{F}} := \left\{(s, a) \mapsto \tilde{W}z(s, a) : \tilde{W} \in \mathbb{R}^{d' \times r}, \|\tilde{W}\|_F \leq L\right\}.$$

By Example 5.8 in Section 5 of Wainwright [2019], we have that

$$\log \mathcal{N}(\tilde{\mathcal{F}}, \epsilon) \leq rd' \log\left(1 + \frac{2L}{\epsilon}\right).$$

This completes the proof. $\qquad\square$

**Lemma D.2.** *With probability at least $1 - \delta$, we have*

$$\mathcal{E}_2(\{P_h^\star\}_{h=0}^{H-1}; \mathcal{D}) = \frac{1}{N} \sum_{n=1}^{N} \sum_{h=0}^{H-1} \sum_{i=0}^{1} \left\| P_h^\star(s_{h+1}^{n,i} | s_h^{n,i}, a_h^{n,i}) - \widehat{P}_h(s_{h+1}^{n,i} | s_h^{n,i}, a_h^{n,i}) \right\|$$

$$\leq \frac{c_1 \xi}{\zeta}\left(H\lambda_{\Sigma_n^{-1}}\sqrt{\frac{\log(2H\mathbb{N}_\mathcal{P}/\delta)}{N}} + H\Gamma(N, \delta)\right),$$

*where* $\xi = c_1\sqrt{\|W^*\|_2 + d'\operatorname{rank}(\Sigma_\mu)\{\operatorname{rank}(\Sigma_\mu) + \log(c_2/\delta)\}\log(1+N)}$, $\Gamma(N, \delta) = \sqrt{\frac{\operatorname{rank}[\Sigma_\mu]\{\operatorname{rank}[\Sigma_\mu] + \ln(c_2/\delta)\}}{N}}$, $c_1$ *and* $c_2$ *are universal constants.*

***Proof of Lemma D.2.*** By the definition of the regularization term, we have

$$\mathcal{E}_2(\{P_h^\star\}_{h=0}^{H-1}; \mathcal{D}) = \frac{1}{N} \sum_{n=1}^{N} \sum_{h=0}^{H-1} \sum_{i=0}^{1} \left\| P_h^\star(s_{h+1}^{n,i} | s_h^{n,i}, a_h^{n,i}) - \widehat{P}_h(s_{h+1}^{n,i} | s_h^{n,i}, a_h^{n,i}) \right\|.$$

Substitute the KNR matrix into the equation above

$$\mathcal{E}_2(\{P_h^\star\}_{h=0}^{H-1}; \mathcal{D}) \leq \frac{1}{N\zeta} \sum_{n=1}^{N} \sum_{h=0}^{H-1} \sum_{i=0}^{1} \|(W^\star - \widehat{W})\phi(s_h^{n,i}, a_h^{n,i})\|_2$$

$$\leq \frac{1}{N\zeta} \sum_{n=1}^{N} \sum_{h=0}^{H-1} \sum_{i=0}^{1} \left\| (W^{\star} - \widehat{W}) \Sigma_n^{1/2} \right\|_2 \cdot \left\| \phi(s_h^{n,i}, a_h^{n,i}) \right\|_{\Sigma_n^{-1}}$$

$$\leq \frac{\xi}{N\zeta} \sum_{n=1}^{N} \sum_{h=0}^{H-1} \sum_{i=0}^{1} \left\| \phi(s_h^{n,i}, a_h^{n,i}) \right\|_{\Sigma_n^{-1}},$$

where the first step is by Lemma 13 in Uehara & Sun [2021b], the second step is by Cauchy-Schwarz inequality, and the last step is by Lemma 12 in Uehara & Sun [2021b], i.e.,

$$\left\| \left( W^{\star} - \widehat{W} \right) (\Sigma_n)^{1/2} \right\|_2 \leq c_1 \sqrt{\|W^*\|_2 + d' \operatorname{rank}(\Sigma_\mu)\{\operatorname{rank}(\Sigma_\mu) + \log(c_2/\delta)\} \log(1+N)} = \xi.$$

We assume that: $Z_\phi := \sum_{h=0}^{H-1} \sum_{i=0}^{1} \left\| \phi(s_h^{n,i}, a_h^{n,i}) \right\|_{\Sigma_n^{-1}}$, so we have

$$\mathcal{E}_2(\{P_h^\star\}_{h=0}^{H-1}; \mathcal{D}) \leq \frac{\xi}{\zeta} \frac{1}{N} \sum_{n=1}^{N} Z_\phi.$$

We also assume: Each $\phi(s,a) \in R^d$ satisfies $\|\phi(s,a)\|_2 \leq 1$, so we have

$$\|\phi(s,a)\|_{\Sigma_n^{-1}} \leq \sqrt{\lambda_{\max}(\Sigma_n^{-1})} \cdot \|\phi(s,a)\|_2 = \sqrt{\lambda_{\max}(\Sigma_n^{-1})} := \lambda_{\Sigma_n^{-1}}.$$

From theorem 21 in Chang et al. [2021], with probability at least $1 - \delta$, we have

$$\mathbb{E}_{(s,a)\sim\mu}[\|\phi(s,a)\|_{\Sigma_n^{-1}}] \leq c_1 \sqrt{\frac{\operatorname{rank}[\Sigma_\mu]\{\operatorname{rank}[\Sigma_\mu] + \ln(c_2/\delta)\}}{N}} = c_1 \Gamma(N,\delta).$$

By applying Hoeffding's inequality and union bound, we have

$$\mathbb{P}\left( \left| \frac{1}{N} \sum_{n=1}^{N} Z_\phi^{(n)} - \mathbb{E}[Z_\phi] \right| \geq \epsilon \right) \leq 2H\mathbb{N}_\mathcal{P} \cdot \exp\left( -\frac{N\epsilon^2}{2H^2 \lambda_{\Sigma_n^{-1}}^2} \right),$$

$$\epsilon = H\lambda_{\Sigma_n^{-1}} \sqrt{\frac{2\log(2H\mathbb{N}_\mathcal{P}/\delta)}{N}}.$$

By Lemma B.4, we have

$$\mathcal{E}_2(\{P_h^t\}_{h=0}^{H-1}; \mathcal{D})$$
$$\leq \frac{cHR_{\max}M_P^K}{\lambda_2} \sqrt{\frac{\log(H\mathbb{N}_\mathcal{P}/\delta)}{N}} + \frac{c\xi}{\zeta} \left( H\lambda_{\Sigma_n^{-1}} \sqrt{\frac{\log(H\mathbb{N}_\mathcal{P}/\delta)}{N}} + H\Gamma(N,\delta) \right).$$

This completes the proof. $\qquad\qquad\square$

Now we want to modify Lemma B.1 to the KNR setting. Specifically, we only need to modify the first term because that is the constraint of the KNR method, i.e, now we want to bound the term

$$\sum_{h=0}^{H-1} \mathbb{E}_{(s,a)\sim d_h^\pi}[D_{TV}(P_h^t(\cdot|s,a), P_h^\star(\cdot|s,a))].$$

**Lemma D.3.** *Let $\pi$ be an arbitrary policy that belongs to $\Pi$. Then for the transition model $\{P_h^t\}_{h=0}^{H-1}$ selected in Line 5 of Algorithm 1 corresponding to $\pi_t$, with probability at least $1 - \delta$, the following holds*

$$\sum_{h=0}^{H-1} \mathbb{E}_{(s,a)\sim d_h^\pi} \left[ D_{TV}(P_h^t(\cdot|s,a), P_h^\star(\cdot|s,a)) \right]$$
$$\leq cH\mathfrak{C}_P^K(\pi) \left( \frac{R_{\max}M_P^K}{\lambda_2} \sqrt{\frac{\log(H\mathbb{N}_\mathcal{P}/\delta)}{N}} + \frac{\xi}{\zeta} \left( \lambda_{\Sigma_n^{-1}} \sqrt{\frac{\log(H\mathbb{N}_\mathcal{P}/\delta)}{N}} + \Gamma(N,\delta) \right) \right),$$

*where $c > 0$ is a universal constant.*

**Proof of Lemma D.3.** We can start by decomposing the term above

$$\sum_{h=0}^{H-1} \mathbb{E}_{(s,a)\sim d_h^\pi} \left[ D_{TV}(P_h^t(\cdot|s,a), P_h^\star(\cdot|s,a)) \right]$$

$$\leq \mathfrak{C}_P^K(\pi) \sum_{h=0}^{H-1} \mathbb{E}_{(s,a)\sim \mu_h} \left[ D_{TV}(P_h^t(\cdot|s,a), P_h^\star(\cdot|s,a)) \right]$$

$$\leq \mathfrak{C}_P^K(\pi) \sum_{h=0}^{H-1} \left( \mathbb{E}_{(s,a)\sim \mu_h} \left[ \left\| P_h^t(\cdot|s,a) - \widehat{P}_h(\cdot|s,a) \right\|_{TV} \right] \right.$$

$$\left. + \mathbb{E}_{(s,a)\sim \mu_h} \left[ \left\| P_h^\star(\cdot|s,a) - \widehat{P}_h(\cdot|s,a) \right\|_{TV} \right] \right).$$

We already have the bound for the second term, so what we need to do now is to achieve a bound for the first term. By Hoeffding's inequality and combined with Lemma D.2, we have

$$\mathbb{E}_{(s,a)\sim \mu_h} \left[ \left\| P_h^t(\cdot|s,a) - \widehat{P}_h(\cdot|s,a) \right\|_{TV} \right]$$

$$\leq \frac{cR_{\max}M_P^K}{\lambda_2} \sqrt{\frac{\log(H\mathbb{N}_\mathcal{P}/\delta)}{N}} + \frac{c\xi}{\zeta} \left( \lambda_{\Sigma_n^{-1}} \sqrt{\frac{\log(H\mathbb{N}_\mathcal{P}/\delta)}{N}} + \Gamma(N,\delta) \right),$$

where $M_P^K = \max_{t\in[1:T]} \sup_{x\in\mathbb{R}^d} \left( \frac{x^\top \Sigma_{\pi_t} x}{x^\top \Sigma_\mu x} \right)$. This completes the proof.

$\square$

Here is the proof of Theorem 5.1.

**Proof of Corollary 5.1.**

$$J(\pi) - J(\pi^{\text{ALG}})$$

$$= \frac{1}{T} \sum_{t=1}^{T} (J(\pi) - J(\pi_t))$$

$$\lesssim \frac{1}{T} \sum_{t=1}^{T} (J_{\mathcal{M}_t}(\pi) - J_{\mathcal{M}_t}(\pi_t)) + \lambda_1 \mathcal{E}_1(r^\star;\mathcal{D}) + \lambda_2 \mathcal{E}_2(\{P_h^\star\}_{h=0}^{H-1};\mathcal{D})$$

$$+ \mathfrak{C}_R(\pi) \sqrt{\frac{\kappa^2 \log(\mathcal{N}_\mathcal{R}(1/N)/\delta)}{N}}$$

$$+ HR_{\max}\mathfrak{C}_P^K(\pi) \left( \frac{R_{\max}M_P^K}{\lambda_2} \sqrt{\frac{\log(H\mathbb{N}_\mathcal{P}/\delta)}{N}} + \frac{\xi}{\zeta} \left( \lambda_{\Sigma_n^{-1}} \sqrt{\frac{\log(H\mathbb{N}_\mathcal{P}/\delta)}{N}} + \Gamma(N,\delta) \right) \right)$$

$$\lesssim R_{\max} \sqrt{\frac{\log|\mathcal{A}|}{T}} + \lambda_1 \mathcal{E}_1(r^\star;\mathcal{D}) + \lambda_2 \mathcal{E}_2(\{P_h^\star\}_{h=0}^{H-1};\mathcal{D})$$

$$+ \mathfrak{C}_R(\pi) \sqrt{\frac{\kappa^2 \log(\mathcal{N}_\mathcal{R}(1/N)/\delta)}{N}}$$

$$+ HR_{\max}\mathfrak{C}_P^K(\pi) \left( \frac{R_{\max}M_P^K}{\lambda_2} \sqrt{\frac{\log(2H\mathbb{N}_\mathcal{P}/\delta)}{N}} + \frac{\xi}{\zeta} \left( \lambda_{\Sigma_n^{-1}} \sqrt{\frac{\log(2H\mathbb{N}_\mathcal{P}/\delta)}{N}} + \Gamma(N,\delta) \right) \right)$$

$$\lesssim R_{\max} \sqrt{\frac{\log|\mathcal{A}|}{T}} + \lambda_1 \left( \sqrt{\frac{\kappa^2 \log(\mathbb{N}_\mathcal{R}/\delta)}{N}} + R_{\max} \sqrt{\frac{\log(2\mathbb{N}_\mathcal{R}/\delta)}{2N}} \right)$$

$$+ \lambda_2 \frac{\xi}{\zeta} \left( H\lambda_{\Sigma_n^{-1}} \sqrt{\frac{\log(H\mathbb{N}_\mathcal{P}/\delta)}{N}} + H\Gamma(N,\delta) \right)$$

$$+ \mathfrak{C}_R(\pi)\sqrt{\frac{\kappa^2 \log(\mathcal{N}_\mathcal{R}(1/N)/\delta)}{N}}$$

$$+ HR_{\max}\mathfrak{C}_P^K(\pi)\left(\frac{R_{\max}M_P^K}{\lambda_2}\sqrt{\frac{\log(H\mathbb{N}_\mathcal{P}/\delta)}{N}} + \frac{\xi}{\zeta}\left(\lambda_{\Sigma_n^{-1}}\sqrt{\frac{\log(H\mathbb{N}_\mathcal{P}/\delta)}{N}} + \Gamma(N,\delta)\right)\right),$$

where the second step is by Lemma B.7, Lemma B.1, the third step is by Lemma C.4, Lemma B.8, Lemma D.3, and The last step is by Lemma D.2, Lemma B.5. Substituting $\lambda_1 = \mathcal{O}(\mathfrak{C}_R(\pi))$, $\lambda_2 = \mathcal{O}(R_{\max}\sqrt{\mathfrak{C}_P^K(\pi)M_P^K})$ completes the proof. $\qquad\square$

## E  TECHNICAL PROOFS FOR FACTORED MODELS

To begin with, we will introduce some notations that will be used in the proofs that follow.

**Summarization of Notations:**

- Number of parameters for the transition functions $L_p = \sum_{i=1}^d |\mathcal{A}| \cdot |\mathcal{B}|^{1+|\mathscr{P}_i|}$.
- Modified Concentrability Coefficient $\mathfrak{C}_P^F(\pi) = \max_{i\in[1:d]} \mathbb{E}_{(s,a)\sim\mu}\left[\left(\frac{d^\pi(s[\mathscr{P}_i],a)}{\mu(s[\mathscr{P}_i],a)}\right)^2\right]$.

**Lemma E.1.** *Let $P_h^\star$ be the ground-truth transition dynamics at time steps $h \in [0:H-1]$, with probability at least $1-\delta$, we have*

$$\mathcal{E}_2(\{P_h^\star\}_{h=0}^{H-1}; \mathcal{D}) = \frac{1}{N}\sum_{n=1}^N \sum_{h=0}^{H-1}\sum_{i=0}^1 \left\|P_h^\star(s_{h+1}^{n,i}|s_h^{n,i},a_h^{n,i}) - \widehat{P}_h(s_{h+1}^{n,i}|s_h^{n,i},a_h^{n,i})\right\|$$

$$\le cH\sqrt{\frac{\log(HdL/\delta)}{N}} + cH\sqrt{\frac{dL\log(LNd/\delta)}{N}},$$

*where $c > 0$ is a universal constant.*

***Proof of Lemma E.1.*** To start with, the following inequality holds

$$\sum_{h=0}^{H-1} \mathbb{E}_{(s,a)\sim\mu_h}[D_{TV}(\widehat{P}_h(\cdot|s,a), P_h^\star(\cdot|s,a))]$$

$$= \sum_{h=0}^{H-1} \mathbb{E}_{(s,a)\sim\mu_h}[\sum_i D_{TV}(\widehat{P}_{i,h}(\cdot|s[\mathscr{P}_i],a), P_{i,h}^\star(\cdot|s[\mathscr{P}_i],a))]$$

$$\le \sum_{h=0}^{H-1}\sum_i \sqrt{\mathbb{E}_{(s,a)\sim\mu}[D_{TV}(\widehat{P}_{i,h}(\cdot|s[\mathscr{P}_i],a), P_{i,h}^\star(\cdot|s[\mathscr{P}_i],a))^2]}$$

$$\le cH\sqrt{\frac{dL\log(LNd/\delta)}{N}},$$

where the second and third steps are by the definition of Factored models, and the last inequality is adapted from section C.5 in Uehara & Sun [2021b].

Define: $Y_n := \sum_{h=0}^{H-1}\sum_{i=0}^1 \left\|P_h^\star(\cdot|s_h^{n,i},a_h^{n,i}) - \widehat{P}_h(\cdot|s_h^{n,i},a_h^{n,i})\right\| \in [0, 4H]$.

By Hoeffding's inequality and apply union bound, we have

$$\Pr\left(\left|\frac{1}{N}\sum_{n=1}^N Y_n - \mathbb{E}[Y_n]\right| \ge \epsilon\right) \le 2HdL\exp\left(\frac{-2N\epsilon^2}{(4H)^2}\right) \Rightarrow \epsilon = 4H\sqrt{\frac{\log(2HdL/\delta)}{2N}}.$$

Then with probability at least $1-\delta$

$$\mathcal{E}_2(\{P_h^\star\}_{h=0}^{H-1}; \mathcal{D}) \le cH\sqrt{\frac{\log(HdL/\delta)}{N}} + cH\sqrt{\frac{dL\log(LNd/\delta)}{N}}.$$

Then by Lemma $B.4$, we have

$$\mathcal{E}_2(\{P_h^t\}_{h=0}^{H-1}; \mathcal{D})$$
$$\lesssim \frac{HM_P^F R_{\max}}{\lambda_2}\sqrt{\frac{\log(HdL/\delta)}{N}} + H\sqrt{\frac{\log(HdL/\delta)}{N}} + H\sqrt{\frac{dL\log(LNd/\delta)}{N}}.$$

This completes the proof. $\qquad\square$

**Lemma E.2.** *Let $\pi$ be an arbitrary policy that belongs to $\Pi$. Then, for any transition dynamics $\{P_h^t\}_{h=0}^{H-1}$ selected in Line 5 of Algorithm 1 corresponding to $\pi_t$, with probability at least $1 - \delta$, the following holds*

$$\sum_{h=0}^{H-1} \mathbb{E}_{d_h^\pi}\left[D_{TV}(P_h^t(\cdot|s,a), P_h^\star(\cdot|s,a))\right]$$
$$\leq cH\mathfrak{C}_P^F(\pi)\left(\frac{R_{\max}M_P^F}{\lambda_2}\sqrt{\frac{\log(HdL/\delta)}{N}} + \sqrt{\frac{\log(HdL/\delta)}{N}} + \sqrt{\frac{dL\log(LNd/\delta)}{N}}\right),$$

*where $c > 0$ is universal constant.*

***Proof of Lemma E.2.*** We start by decomposing the term above

$$\sum_{h=0}^{H-1} \mathbb{E}_{d_h^\pi}\left[D_{TV}(P_h^t(\cdot|s,a), P_h^\star(\cdot|s,a))\right]$$
$$\leq \mathfrak{C}_P^F(\pi)\sum_{h=0}^{H-1}\left(\mathbb{E}_{(s,a)\sim\mu_h}\left[\left\|P_h^t(\cdot|s,a) - \widehat{P}_h(\cdot|s,a)\right\|_{TV}\right]\right.$$
$$\left. + \mathbb{E}_{(s,a)\sim\mu_h}\left[\left\|P_h^*(\cdot|s,a) - \widehat{P}_h(\cdot|s,a)\right\|_{TV}\right]\right),$$

where $\mathfrak{C}_P^F(\pi) = \max_{i\in[1:d]}\mathbb{E}_{(s,a)\sim\mu}\left[\left(\frac{d^\pi(s[\mathscr{P}_i],a)}{\mu(s[\mathscr{P}_i],a)}\right)^2\right]$.

By Hoeffding's inequality and combined with Lemma E.1

$$\mathbb{E}_{(s,a)\sim\mu_h}\left[\left\|P_h^t(\cdot|s,a) - \widehat{P}_h(\cdot|s,a)\right\|_{TV}\right]$$
$$\leq \frac{cM_P^F}{\lambda_2}\sqrt{\frac{\log(HdL/\delta)}{N}} + c\sqrt{\frac{\log(HdL/\delta)}{N}} + c\sqrt{\frac{dL\log(LNd/\delta)}{N}},$$

where $M_P^F = \max_{t\in[1:T]}\max_{i\in[1:d]}\mathbb{E}_{(s,a)\sim\mu}\left[\left(\frac{d^{\pi_t}(s[\mathscr{P}_i],a)}{\mu(s[\mathscr{P}_i],a)}\right)^2\right]$. Combining all the terms above, we have

$$\mathbb{E}_{(s,a)\sim d_h^\pi}\left[\left\|P_h^t(\cdot|s,a) - P_h^\star(\cdot|s,a)\right\|_{TV}\right]$$
$$\leq \mathfrak{C}_P^F(\pi)\left(\frac{cR_{\max}M_P^F}{\lambda_2}\sqrt{\frac{\log(HdL/\delta)}{N}} + c\sqrt{\frac{\log(HdL/\delta)}{N}} + c\sqrt{\frac{dL\log(LNd/\delta)}{N}}\right).$$

This completes the proof. $\qquad\square$

Now we present the proof of the PAC bound for Factored Models.

***Proof of Theorem 5.2.***

$$J(\pi) - J(\pi^{\text{ALG}})$$
$$= \frac{1}{T}\sum_{t=1}^{T}(J(\pi) - J(\pi_t))$$

$$\lesssim \frac{1}{T}\sum_{t=1}^{T}\left(J_{\mathcal{M}_t}(\pi) - J_{\mathcal{M}_t}(\pi_t)\right) + \lambda_1 \mathcal{E}_1(r^\star; \mathcal{D}) + \lambda_2 \mathcal{E}_2(\{P_h^\star\}_{h=0}^{H-1}; \mathcal{D})$$

$$+ H R_{\max}\mathfrak{C}_P^F(\pi)\left(\frac{R_{\max}M_P^F}{\lambda_2}\sqrt{\frac{\log(HdL/\delta)}{N}} + \sqrt{\frac{\log(HdL/\delta)}{N}} + \sqrt{\frac{dL\log(LNd/\delta)}{N}}\right)$$

$$+ \mathfrak{C}_R(\pi)\sqrt{\frac{c\kappa^2\log(rL_1/\delta)}{N}}$$

$$\lesssim R_{\max}\sqrt{\frac{\log|\mathcal{A}|}{T}} + \lambda_1 \mathcal{E}_1(r^\star; \mathcal{D}) + \lambda_2 \mathcal{E}_2(\{P_h^\star\}_{h=0}^{H-1}; \mathcal{D})$$

$$+ H R_{\max}\mathfrak{C}_P^F(\pi)\left(\frac{R_{\max}M_P^F}{\lambda_2}\sqrt{\frac{\log(HdL/\delta)}{N}} + \sqrt{\frac{\log(HdL/\delta)}{N}} + \sqrt{d\frac{L\log(LNd/\delta)}{N}}\right)$$

$$+ \mathfrak{C}_R(\pi)\sqrt{\frac{\kappa^2\log(rL_1/\delta)}{N}}$$

$$\lesssim R_{\max}\sqrt{\frac{\log|\mathcal{A}|}{T}} + \lambda_1\left(\sqrt{\frac{\kappa^2\log(rL/\delta)}{N}} + R_{\max}\sqrt{\frac{\log(rL/\delta)}{N}}\right)$$

$$+ \lambda_2\left(H\sqrt{\frac{\log(dL/\delta)}{N}} + H\sqrt{\frac{dL\log(LNd/\delta)}{N}}\right)$$

$$+ H R_{\max}\mathfrak{C}_P^F(\pi)\left(\frac{R_{\max}M_P^F}{\lambda_2}\sqrt{\frac{\log(HdL/\delta)}{N}} + \sqrt{\frac{\log(HdL/\delta)}{N}} + \sqrt{\frac{dL\log(LNd/\delta)}{N}}\right)$$

$$+ \mathfrak{C}_R(\pi)\sqrt{\frac{\kappa^2\log(rL_1/\delta)}{N}},$$

where the second step is by Lemma B.7, Lemma B.1, Lemma B.8, Lemma E.2, the third step is by Lemma C.4, and the last step is by Lemma B.5, Lemma E.1. Substituting $\lambda_1 = \mathcal{O}(\mathfrak{C}_R(\pi))$, $\lambda_2 = \mathcal{O}(R_{\max}\sqrt{\mathfrak{C}_P^F(\pi)M_P^F})$ completes the proof. $\qquad\square$

# F  EXPERIMENTAL SETUP

## F.1  DATASETS

Our method is assessed using the Meta-World datasets introduced by Yu et al. [2020a], specifically the medium-replay curated by Choi et al. [2024]. The primary set of experiments focuses on the medium-replay dataset. A key advantage of these datasets is that policies cannot achieve good performance when trained with incorrect reward signals (e.g., random or constant), making them particularly well-suited for evaluating offline reinforcement learning approaches. This is important because some offline RL agents may still exhibit seemingly successful behavior even under faulty reward supervision. Further dataset-related information is available in Choi et al. [2024].

Note that the abbreviation *BPT* indicates *button-press-topdown*.

The medium-replay dataset used in our study, originally introduced by Choi et al. [2024], comprises replay buffers created using SAC agents [Haarnoja et al., 2018], which exhibit an average success performance around 50%.

## F.2  COMPUTATIONAL RESOURCES

Our experiments are conducted on a single Nvidia GeForce RTX 5090 GPU. Each training session consists of 100,000 gradient steps, taking approximately 50 minutes to complete (with evaluation).

# G    IMPLEMENTATION DETAILS

## G.1    EXPERIMENTAL DETAIL OF BENCHMARKS

The Table 4 provides a comprehensive summary of the reward model configuration used across benchmark implementations, along with the key experimental settings for algorithms including IQL (represented as Oracle in the table), MR, PT, DPPO, IPL, and APPO. These settings cover essential implementation details such as network architecture, optimizer type, learning rates, batch sizes, and the number of training epochs, ensuring transparency and fairness in reproduction and comparison.

Table 4: Implementation details of the reward model and the baselines.

| Algorithm | Component | Value |
|---|---|---|
| **Reward model** | Optimizer | Adam Kingma & Ba [2014] |
| | Learning rate | 1e-3 |
| | Batch size | 512 |
| | $Q$ | 100 |
| | Hidden layer dim | 128 |
| | Hidden layers | 3 |
| | Activation function | ReLU |
| | Final activation | Tanh |
| | Epochs | 300 |
| | # of ensembles | 3 |
| | Reward from the ensemble models | Average |
| **IQL  Kostrikov et al. [2021]** | Optimizer | Adam Kingma & Ba [2014] |
| | Critic, Actor, Value hidden dim | 256 |
| | Critic, Actor, Value hidden layers | 2 |
| | Critic, Actor, Value activation function | ReLU |
| | Critic, Actor, Value learning rate | 0.5 |
| | Mini-batch size | 256 |
| | Discount factor | 0.99 |
| | $\beta$ | 3.0 |
| | $\tau$ | 0.7 |
| **MR  Lee et al. [2021]** | Neural networks ($Q, V, \pi$) | 3-layers, hidden dimension 256 |
| | Activation | ReLU for hidden activations |
| | $Q, V, \pi$ optimizer | Adam with learning rate 3e-4 |
| | Batch size | 256 |
| | Target network soft update | 0.005 |
| | $\beta$ (IQL advantage weight) | 3.0 |
| | $\tau$ (IQL expectile parameter) | 0.7 |
| | Discount factor | 0.99 |
| **PT  Kim et al. [2023]** | Optimizer | AdamW Loshchilov & Hutter [2017] |
| | # of layers | 1 |
| | # of attention heads | 4 |
| | Embedding dimension | 256 |
| | Dropout rate | 0.1 |
| **DPPO  An et al. [2023]** | Preference predictor | The same as PT Kim et al. [2023] |
| | Smoothness regularization $\nu$ | 1.0 |
| | Smoothness sigma $m$ | 20 |
| | Regularization $\lambda$ | 0.5 |
| **IPL  Hejna & Sadigh [2023]** | Optimizer | Adam Kingma & Ba [2014] |
| | Regularization $\lambda$ | 3e-4 |
| | $Q, V, \pi$ arch | 3x256d |
| | $\beta$ | 4.0 |
| | $\tau$ | 0.7 |
| | Subsample $s$ | 16 |
| **APPO  Kang & Oh [2025]** | Neural networks ($Q, V, \pi$) | 3-layers, hidden dimension 256 |
| | Activation | LeakyReLU for hidden activations |
| | $Q, V, \alpha$ optimizer | Adam with learning rate 3e-4 |
| | $\pi$ optimizer | Adam with learning rate 3e-5 |
| | Batch size | 256 transitions and 16 trajectory pairs |
| | Target network soft update | 0.001 |
| | Discount factor | 0.99 |

## G.2 BASIC EXPERIMENTAL DESIGN

The preference dataset is constructed by randomly drawing pairs of trajectory fragments, each consisting of 25 time steps. Preference labels are assigned using the true reward: a label of (0,1) is used if the total rewards of the two trajectories differ by more than 12.5; otherwise, both segments receive an equal preference label of (0.5, 0.5). Performance is quantified by task-specific success rates, which reflect whether the agent manages to complete the intended task.

## G.3 TRAINING DETAILS

In the following, we talk about the details of the model implementation. We firstly represent the dynamics model as an ensemble of neural networks that output a Gaussian distribution over the next state given the current state and action:

$$\hat{P}(s' \mid s, a) = \mathcal{N}\big(\mu(s,a), \Sigma(s,a)\big).$$

Following previous works Rigter et al. [2022]; Sun [2023], during the initial maximum likelihood model training (Line 3 of Algorithm 2) we train an ensemble of 7 such dynamics models and pick the best 5 models based on the validation error on a held-out test set of 10000 transitions from the offline dataset $\mathcal{D}$. Each model in the ensemble is a 4-layer feedforward neural network with 200 hidden units per layer. We summarize the details of the algorithm in Algorithm 2.

---

**Algorithm 2** **M**odel-based **C**onservative **P**lanning (MCP)

---

**Input:** Offline dataset $\mathcal{D}$; regularization parameters $\lambda_1, \lambda_2, \lambda_b$; rollout length $l$; sub-trajectory length $h$:

1: Initialize the policy $\pi_1$ as the uniform policy.
2: **Learn reward:** $\hat{r} = \arg\max_{r \in \mathcal{R}} \sum_{n=1}^{N} \log P_r(o = o^n \mid \tau^{n,1}, \tau^{n,0})$.
3: **Learn transition kernel:** for all $h \in \{0, \dots, H-1\}$,

$$\widehat{P}_h = \underset{P_h \in \mathcal{P}_h}{\arg\max} \sum_{n=1}^{N} \sum_{i=0}^{1} \log P_h(s_{h+1}^{n,i} \mid s_h^{n,i}, a_h^{n,i}).$$

4: **for** $t = 1, 2, \dots, T$ **do**
**(a) Model update:**
  (i) Initialize the reward model $r$ and transition kernel $\{P_h\}_{h=0}^{H-1}$ with the MLE models $\hat{r}$ and $\{\widehat{P}_h\}_{h=0}^{H-1}$.
  (ii) Collect a sub-trajectory $\tau^{\mathrm{sub}}$ from $s_{\mathrm{sub}}$ of length $h$ and compute $r(\tau^{\mathrm{sub}})$.
  (iii) Roll out $\pi_t$ from $s_{\mathrm{sub}}$ under $\{P_h \in \mathcal{P}_h\}_{h=0}^{H-1}$ for $l$ steps to obtain $\tau^{\mathrm{eval}}$.
  (iv) Solve for $r_t$ and $\{P_h^t\}_{h=0}^{H-1}$:

$$\min_{r, \{P_h\}} \mathbb{E}_{d_{\{P_h\}_{h=0}^{H-1}}^{\pi_t}} \big[r(\tau^{\mathrm{eval}}) \mid s_{\mathrm{sub}}\big] - r(\tau^{\mathrm{sub}}) + \lambda_1 \, \mathcal{E}_1(r; \mathcal{D}) + \lambda_2 \, \mathcal{E}_2\big(\{P_h\}_{h=0}^{H-1}; \mathcal{D}\big).$$

**(b) Policy update:**
  (i) Initialize $\pi \leftarrow \pi_t$.
  (ii) Sample a batch of $(s, a)$ from $\mathcal{D}$ and draw $a' \sim \pi_t(\cdot \mid s)$.
  (iii) Obtain $\pi_t^{\mathrm{loc}}$ by

$$\arg\max_{\pi} \Big\langle \pi(\cdot \mid s), \mathbb{E}_{d_{\{P_h^t\}_{h=0}^{H-1}}^{\pi_t}}\big[r_t(\tau) \mid s, \cdot\big] \Big\rangle - \lambda_b \, (a' - a)^2.$$

  (iv) Align $\pi_t^{\mathrm{loc}}(\cdot \mid s)$ with $\pi_t(\cdot \mid s)$ by minimizing the KL divergence to obtain $\pi_{t+1}$.
5: **end for**
6: **Output:** $\pi^{\mathrm{ALG}} = \pi^T$.

---

In (a)(iii), we follow TD3-BC [Fujimoto & Gu, 2021] to include a regression–style behavior cloning loss to push the policy towards favoring actions contained in the dataset. In (b)(iv), we leverage the subtle decremental training for KL-divergence trust region updating to avoid $\pi_t^{\mathrm{loc}}$ getting too close to $\pi_t$. To ensure a fair comparison with baseline methods, we follow the official implementation

provided by Choi et al. [2024] for training our reward model. Specifically, the reward model is implemented as an ensemble of three fully connected neural networks, each consisting of three hidden layers with 128 units per layer. ReLU is used as the activation function for all hidden layers, and a Tanh activation is applied to the final output. The ensemble prediction is computed as the average of the individual network outputs.

Table 5: Model architecture and hyperparameters.

|  | Component | Value |
|---|---|---|
| Our work | Neural networks $(r, P, \pi)$ | 3-layers, hidden dimension 256 |
|  | Activation | ReLU for hidden activations |
|  | $\pi$ optimizer | Adam with learning rate 1e-4 |
|  | Batch size | 256 transitions and 256 trajectory pairs |
|  | $\lambda_1$ | $1^\star$ |
|  | $\lambda_2$ | $0.01^\star$ |
|  | $\lambda_b$ | $1^\star$ |
|  | Rollout length $l$ | $2^\star$ |
|  | Sub-trajectory length $h$ | $5^\star$ |
|  | no. of model networks | 7 |
|  | no. of elites | 5 |
|  | model learning rate | 3e-4 |

$\star$ The data marked with asterisks are only approximate values and need to be adjusted according to different datasets.

## H  VISUALIZATION OF TRAINING DYNAMICS

Figure 3 illustrates the training progress corresponding to the experiments summarized in Table 2. Each algorithm undergoes training for a total of 100,000 gradient update steps, during which performance evaluations are performed at regular intervals of every 5,000 steps. To derive the final success rate reported in the tables, we compute the average and standard deviation over the results from three random seeds, offering a more stable and representative performance estimate. This figure is generated from the mean, standard derivation of the success rates.

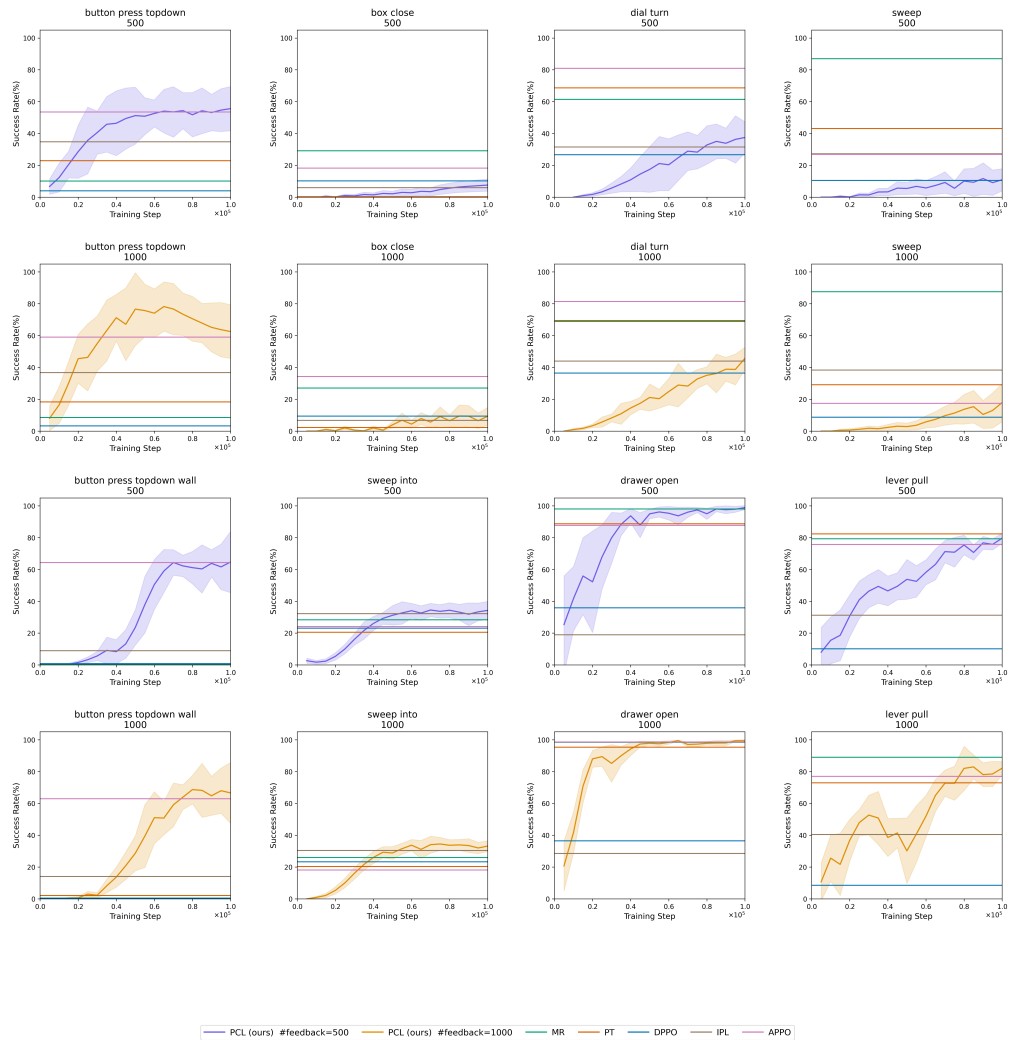

Figure 3: Learning curves from the experiments in Table 2 of main text.

## I  THE USE OF LLM

The authors used a large language model (LLM) to edit and polish the writing (grammar, wording, and clarity). All ideas, analyses, and conclusions are the authors' own.

## J  ADDITIONAL EXPERIMENTAL RESULTS

**Generalization to high-dimensional environments.**  To evaluate the performance of MCP in more complex environments with significantly larger state–action spaces, we construct new offline

PbRL datasets based on the MuJoCo tasks Ant [Schulman et al., 2015] and Humanoid [Tassa et al., 2012]. These two tasks have high-dimensional states and actions: Ant has a state space of dimension 105 (about 3 times larger than MetaWorld) and an action space of dimension 8 (about 2 times the action dimension of MetaWorld), while Humanoid has a state space of dimension 348 (about 9 times larger than MetaWorld) and an action space of dimension 17 (about 4 times the action dimension of MetaWorld). Table 6 reports the average total return over three seeds for each method. As shown in Table 6, MCP consistently outperforms MR and APPO on both high-dimensional tasks, demonstrating its potential in complex environments.

Table 6: Performance on high-dimensional MuJoCo tasks.

| Algorithm | Oracle | MR | APPO | MCP |
|---|---|---|---|---|
| Humanoid | 2672.48 | 1586.31 | 1365.34 | 1731.47 |
| Ant | 3007.22 | 1940.87 | 1638.42 | 2213.85 |

**Runtime analysis.** We measure the time required to perform 100k training steps using a single NVIDIA RTX 5090 GPU. As reported in Table 7, MCP is slightly slower than MR but still faster than APPO, while achieving better performance than both. Table 8 further decomposes the MCP runtime into the conservative planning step (line 5 of Algorithm 1) and the policy update step (line 6).

Table 7: Running time for 100k training steps.

| Method | MR | APPO | MCP |
|---|---|---|---|
| Total time (min) | 22.7 | 36.8 | 33.5 |

Table 8: Decomposition of MCP runtime for 100k training steps.

| Method | MCP-step5 | MCP-step6 | MCP-total |
|---|---|---|---|
| Total time (min) | 24.2 | 9.3 | 33.5 |

**Hyperparameter sensitivity.** We conduct a hyperparameter sensitivity analysis for the behavior cloning weights $\lambda_b$ on the *drawer-open-1000* and *sweep-into-1000* tasks over three seeds.

Table 9: Sensitivity of MCP to regularization hyperparameters on *drawer-open-1000* and *sweep-into-1000*.

| $\lambda_b$ | 1e-1 | 3e-1 | 1 | 3 | 1e1 |
|---|---|---|---|---|---|
| drawer-open-1000 | $92.0 \pm 5.66$ | $97.20 \pm 2.04$ | $99.07 \pm 0.38$ | $96.40 \pm 3.81$ | $89.87 \pm 9.92$ |
| sweep-into-1000 | $22.53 \pm 7.39$ | $26.8 \pm 4.25$ | $29.2 \pm 6.76$ | $34.13 \pm 3.27$ | $31.6 \pm 5.02$ |

