# OpenReview forum: "Toward Conservative Planning from Human-AI Preferences in Reinforcement Learning"
_ICLR.cc/2026/Conference — ICLR 2026 Poster_

### Official Review · Reviewer_Ec41 · 2025-10-30

**Soundness:** 2
**Presentation:** 3
**Contribution:** 2
**Rating:** 4
**Confidence:** 4

**Summary:**

In this paper, the authors study offline reinforcement learning (RL) with trajectory preferences, where the RL agent does not receive explicit rewards at each step but instead receives human-provided preferences over pairs of trajectories. They propose a novel Model-based Conservative Planning (MCP) algorithm for offline PbRL, which leverages a general function class and uses a tractable conservative learning framework to improve the policy upon an arbitrary reference policy. They prove that, MCP can compete with the best policy within data coverage when the reference policy is supported by the data. Lastly, the authors conduct some empirical evaluations of MCP.

**Strengths:**

1. The problem of conservative planning in RLHF is an important and interesting problem.
2. The technical part of this paper is solid, the proof looks correct to me.
3. There are simulation results supporting the theoretical results.

**Weaknesses:**

1. My main concern is about the algorithm. The hyperparameter $\lambda_1$ and $\lambda_2$ depend on the concentrability coefficients, which further depends on the real reward or transition kernel. Would you please explain how to calculate (or estimate) such values efficiently?

2. Furthermore, $\lambda_2$ also depends on $M_P$, which is a max over all the intermediate steps during the training. How could the algorithm know this value before the algorithm starts?

3. The idea behind line 5 and 6 of alg.1 is interesting. Would you please analyze the computational complexity of solving the optimization problem in line 5?

**Questions:**

Please see the weakness

---

> ### Author Response · Authors · 2025-11-23
>
> We are grateful for the reviewer's constructive feedback and insightful comments. In the following, we provide point-by-point responses to all your comments, and we hope our responses can help to resolve your concerns.
>
> **W1["My main concern is about the algorithm. The hyperparameter
> $\lambda_1$ and $\lambda_2$ depend on the concentrability coefficients..."]**
>
> We thank the reviewer for this important question. In our theory, the
> regularization weights $\lambda_1$ and $\lambda_2$ are chosen on the order of the concentrability coefficients, but these coefficients depend on the unknown true MDP and are not meant to be computed exactly in practice. In our implementation, we therefore treat $(\lambda_1,\lambda_2)$ as
> regularization hyperparameters on the reward model, the transition model, respectively, and select them by online
> hyperparameter tuning.
>
> The reason this online selection is justified in our setting is precisely our safe policy improvement corollary (Corollary 4.2), which is derived from the relative-performance objective. When the comparator is the behavior policy $\pi_b$, the corollary guarantees that the learned policy
> $\pi_{\text{ALG}}$ is at least as good as $\pi_b$ up to statistical error,
> i.e., $J(\pi_{\text{ALG}}) \gtrsim J(\pi_b)$ with high probability. Thus, when we run MCP with different $(\lambda_1,\lambda_2)$ and evaluate the resulting policies online, every candidate is already guaranteed to be essentially no worse than the behavior policy, so choosing the best-performing one among them does not violate our safety guarantee.
>
> To further support that MCP does not require delicate tuning of these
> regularization weights, we have added a sensitivity analysis for
> $(\lambda_1,\lambda_2)$ on the drawer-open-1000 and
> sweep-into-1000 tasks. As shown in the table below, the success rates for different values of each
> hyperparameters (while keeping the others fixed) vary only mildly, demonstrating that MCP is robust to the
> choice of these regularization coefficients.
>
>
> drawer-open:
> | $\lambda_1$ | 1e-1   | 3e-1       |     1       | 3      | 1e1 |
> |------------|-------|------------|-------------|--------|--------|
> |success rate|94.8±3.85|97.60±2.33|99.07±0.38|94.53±4.03|86.27±10.31|
>
> | $\lambda_2$ | 3e-3   | 1e-2       |     3e-2       |  1e-1     | 1 |
> |------------|-------|------------|-------------|--------|--------|
> | success rate|90.93±7.3|95.47±4.10|99.07±0.38|89.6±9.53|80.13±15.19|
>
>
>
> sweep-into:
> | $\lambda_1$ | 1e-1   | 3e-1       |     1       | 3      | 1e1 |
> |------------|-------|------------|-------------|--------|--------|
> |success rate|32.4±5.85|34.13±3.27|30.67±4.71|24±9.27|19.87±7.92|
>
> | $\lambda_2$ | 3e-3   | 1e-2       |     3e-2       |  1e-1     | 1 |
> |------------|-------|------------|-------------|--------|--------|
> | success rate|20.8±10.32|23.6±5.85|28.27±7.08|34.13±3.27|25.73±6.15|
>
>
>
>
>
>
> **W2["Furthermore, $\lambda_2$ also depends on $M_P$, which is a max over all the intermediate steps during the training..."]**
>
> We thank the reviewer for raising this point. Indeed, in our theoretical analysis, the optimal scaling of $\lambda_2$ depends on $M_P$, which is defined as a maximum over the sequence of intermediate policies and models encountered during training. This dependence is only used to characterize the size of the statistical error term and is not intended to be evaluated exactly by the algorithm. In particular, MCP does not need to know $M_P$ in order to run: the bound simply says that if the unknown constant $M_P$ is larger, then the theoretically optimal $\lambda_2$ should be larger as well (i.e., stronger pessimism is required under worse coverage).
>
> In practice, we therefore treat $\lambda_2$ as a regularization hyperparameter that does not need to track $M_P$ explicitly. We select $\lambda_2$ by online evaluation over a small grid of candidate values. This procedure is justified by our safe policy improvement corollary (Corollary 4.2): when the comparator is the behavior policy $\pi_b$, MCP guarantees that the resulting policy $\pi_{\text{ALG}}$ is at least as good as $\pi_b$ up to statistical error, so choosing the best-performing $\lambda_2$ among a set of runs remains compatible with our safety guarantee.
>
> Finally, the hyperparameter sensitivity analysis reported in the tables above further supports that MCP does not require a precise choice of $\lambda_2$. Varying $\lambda_2$ over several orders of magnitude leads to only mild changes in success rate and does not cause catastrophic degradation, indicating that MCP is robust to the exact value of $\lambda_2$ and does not rely on accurately knowing $M_P$ before training.

---

> > ### Author Response · Authors · 2025-11-23
> >
> > **W3["The idea behind line 5 and 6 of alg.1 is interesting. Would you please analyze the computational complexity of solving the optimization problem in line 5?"]**
> >
> > We thank the reviewer for the question and the interest in the complexity of lines 5 and 6 of Algorithm 1. To make this concrete, we measure the time required to run 100k training steps for our method and baselines on the same machine with a single RTX 5090 GPU. The results are reported in Table 1. MCP takes about 33.5 minutes in total, which is slower than MR but still slightly faster than APPO, while at the same time achieving better final performance than both baselines.
> >
> > Table 2 decomposes the MCP runtime into the time spent on line 5 (MCP-step5), which solves the conservative planning problem with the learned model, and line 6 (MCP-step6), which performs the policy update. We observe that line 5 accounts for roughly 72% of the total runtime and line 6 for about 28%. These measurements show that the additional optimization in lines 5 and 6 introduces only a moderate computational overhead and does not make MCP computationally prohibitive compared to existing baselines.
> >
> > Table 1
> > | Method |MR|APPO|MCP|
> > |--------|------------------|-------------------|-------------------|
> > | Total time (min) |22.7|36.8|24.2|9.3|33.5|
> >
> > Table 2
> > | Method |MCP-step5|MCP-step6|MCP|
> > |--------|------------------|-------------------|------------------|
> > | Total time (min) |24.2|9.3|33.5|
> >
> >
> >
> > [1] Preference transformer: Modeling human preferences using transformers for RL (Kim et al., 2023)
> >
> > [2] Adversarial policy optimization for offline preference-based reinforcement learning (Kang et al., 2025)

---

### Official Review · Reviewer_GUzu · 2025-10-31

**Soundness:** 2
**Presentation:** 3
**Contribution:** 2
**Rating:** 4
**Confidence:** 4

**Summary:**

This paper proposes Model-based Conservative Planning (MCP), a framework for offline preference-based reinforcement learning (PbRL) that integrates model learning with conservative optimization to ensure reliable performance under partial data coverage. The authors provide generalization guarantees under general function approximation and extend the analysis to structured settings such as kernelized nonlinear regulators. Empirically, the paper evaluates MCP on the Meta-World medium-replay benchmark, showing improved success rates over several preference-based baselines (PT, IPL, DPPO, APPO).

**Strengths:**

- The paper is clearly written and logically structured.

- The paper provides a solid theoretical framework for offline preference-based reinforcement learning under partial coverage. It establishes generalization guarantees for model-based conservative planning with general function approximation, which, to the best of my knowledge, has not been analyzed in prior PbRL literature.

**Weaknesses:**

The experimental evaluation exhibits certain limitations.

- The experiments are conducted solely on the Meta-World medium-replay benchmark, which primarily contains deterministic, low-dimensional robotic tasks. This restricted evaluation makes it difficult to assess how well the proposed conservative planning approach generalizes to more diverse or high-dimensional offline preference-based RL settings. Including broader datasets such as D4RL or human-feedback benchmarks (e.g., Atari or MuJoCo preference datasets) would strengthen the empirical evidence.

- The paper lacks ablations isolating the contribution of each design component, such as the relative performance regularization, model-based planning, and the choice of regularization weights.  Without these analyses, it is unclear how much improvement stems from the conservative objective itself versus other implementation factors. Including sensitivity studies or component removals would better validate the claimed effectiveness.

**Questions:**

See weaknesses.

---

> ### Author Response · Authors · 2025-11-23
>
> We are grateful for the reviewer's constructive feedback and insightful comments. In the following, we provide point-by-point responses to all your comments.
>
> **W1["The experiments are conducted solely on the Meta-World medium-replay benchmark..."]**
>
> We follow the reviewer's suggestion to evaluate the performance of MCP in complex environments with significantly larger state-action spaces. Due to the absence of existing offline PbRL datasets with extremely high-dimensional states and actions, we consider benchmarking a new set of offline PbRL datasets on our own. In particular, we consider the Mujoco tasks. We conduct experiments on the environments Ant [1] and Humanoid [2]. These two tasks have high-dimensional states and actions: Ant has a state space of dimension $105$ ($3$ times larger than MetaWorld) and an action space of dimension 8($2 $ times the action dimension of MetaWorld), and Humanoid has a state space of dimension $348$ ($9$ times larger than MetaWorld) and an action space of dimension $17$ ($4$ times the action dimension of MetaWorld).
>
> Benchmarking these new offline PbRL datasets is both time-consuming and resource-intensive. Given the time constraints, our experiments focus on comparing MCP with MR [3] and APPO [4], one of the most competitive baselines in offline PbRL. The results are summarized in the following table, where we report the mean of the raw return from three seeds.
>
> |Algorithm|Orcale|MR|APPO|MCP|
> |------------|-------|------------|-------------|-------------|
> |Humanoid|2672.48|1586.31|1365.34|1731.47|
> |Ant|3007.22|1940.87|1638.42|2213.85|
>
> As shown in the table above, MCP consistently outperforms MR and APPO across the two high-dimensional tasks. This demonstrates the potential of MCP in high-dimensional complex environments.
>
> **W2["The paper lacks ablations isolating the contribution of each design component..."]**
>
> We thank the reviewer for emphasizing the importance of ablations on the design components. In the revised manuscript, we add an ablation study on the lever-pull task with 1000 preferences (new Figure 2(a)), where we keep the network architectures, dataset, and optimization settings fixed and modify only the key components of MCP.
>
> Figure 2(a) shows that full MCP rapidly reaches a high success rate and then remains stable, while all ablated variants stay much lower and learn more slowly. Removing either the reward or the transition regularizer ($\lambda_1 = 0$ or $\lambda_2 = 0$) significantly degrades performance, indicating that the conservative terms are necessary to prevent over-optimistic and unstable policies. The “No Relative Performance" variant also underperforms and exhibits large variability, which is consistent with the absence of the safe policy improvement guarantee. Finally, the “Value-based MCP" variant improves briefly in the early stage but then plateaus and eventually degrades. This suggests that replacing the model-based planner with a purely value-based update introduces noticeable bias from model errors. In the preference-based setting, where the reward signal is already noisy and indirect, adding such extra bias can easily destabilize learning, which motivates our use of conservative model-based planning in MCP.
>
> Overall, these ablations confirm that the conservative regularizers, the relative-performance objective, and the model-based planning step all play important roles in the effectiveness of MCP.

---

> > ### Author Response · Authors · 2025-11-23
> >
> > **W2["The paper lacks ablations isolating the contribution of each design component..."]**
> >
> > In addition, we conduct a hyperparameter sensitivity analysis for the regularization weights $\lambda_1, \lambda_2$, corresponding respectively to the reward regularizer and the transition regularizer, on the drawer-open and sweep-into tasks.
> >
> > Drawer-open:
> > |$\lambda_1$| 1e-1   | 3e-1       |     1       | 3      | 1e1 |
> > |------------|-------|------------|-------------|--------|--------|
> > |success rate|94.8±3.85|97.60±2.33|99.07±0.38|94.53±4.03|86.27±10.31|
> >
> > |$\lambda_2$| 3e-3   | 1e-2       |     3e-2       |1e-1 | 1 |
> > |------------|-------|------------|-------------|--------|--------|
> > | success rate|90.93±7.3|95.47±4.10|99.07±0.38|89.6±9.53|80.13±15.19|
> >
> > Sweep-Into:
> > | $\lambda_1$ | 1e-1   | 3e-1       |     1       | 3      | 1e1 |
> > |------------|-------|------------|-------------|--------|--------|
> > |success rate|32.4±5.85|34.13±3.27|30.67±4.71|24±9.27|19.87±7.92|
> >
> > | $\lambda_2$| 3e-3   | 1e-2       |     3e-2       |  1e-1     | 1 |
> > |------------|-------|------------|-------------|--------|--------|
> > | success rate|20.8±10.32|23.6±5.85|28.27±7.08|34.13±3.27|25.73±6.15|
> >
> >
> >
> > As the table above, across several orders of magnitude for each parameter, the success rate of MCP changes only mildly, showing that our method is not highly sensitive to the exact choice of $(\lambda_1, \lambda_2)$. Together, these ablations and sensitivity studies provide direct evidence that each proposed design component contributes meaningfully to performance, and that the effectiveness of MCP does not rely on delicate hyperparameter tuning.
> >
> >
> > [1] High-dimensional continuous control using generalized advantage estimation (Schulman et al., 2015)
> >
> > [2] Synthesis and stabilization of complex behaviors through online trajectory optimization (Tassa et al., 2012)
> >
> > [3] Preference transformer: Modeling human preferences using transformers for RL (Kim et al., 2023)
> >
> > [4] Adversarial policy optimization for offline preference-based reinforcement learning (Kang et al., 2025)

---

### Official Review · Reviewer_6XMk · 2025-11-01

**Soundness:** 3
**Presentation:** 2
**Contribution:** 3
**Rating:** 4
**Confidence:** 3

**Summary:**

This work proposes MCP, a novel model-based conservative planning algorithm for offline preference-based RL that is both sample-efficient and computationally tractable. Under partial coverage, MCP does not require additional structural assumptions and provides a PAC guarantee with a tractable implementation in model-based RL. To the best of the reviewer’s knowledge, this is the first work in model-based RL that addresses the limitations of PbRL. In experiments, MCP demonstrates high performance compared to other baselines.

**Strengths:**

1. The authors derive PAC bounds for the three variants of MCP (original, factored, and KNR). The paper is clearly structured, making it easy for readers to follow the overall flow and reasoning.

2. This work is a natural extension to the model-based setting and effectively addresses the limitations of previous approaches.

**Weaknesses:**

1. In Figure 1-(a), MCP is compared only with MR and Oracle. It would strengthen the validity of the theoretical claims if the experiments also included a comparison with APPO, as was done in the theoretical analysis regarding sample efficiency (Line 310).

2. The training curves are presented only for MCP. It would be beneficial to include those of other baselines as well, so that the results highlight not only the final performance but also the faster convergence of MCP, demonstrating the advantage of the model-based RL approach.

**Questions:**

1. The authors mention that MCP provides an implicit way of encoding conservatism, mainly through the minimax objective function. However, it is somewhat difficult to intuitively understand how this mechanism works in practice. Could the authors provide a simple or toy example to illustrate how the minimax structure implicitly enforces conservatism?

2. In Table 5, $\lambda_{1}$, $\lambda_{2}$, and $\lambda_{3}$ are important parameters for MCP. Therefore, it would be helpful to report their exact values for each dataset. Are these parameters highly sensitive?

---

> ### Author Response · Authors · 2025-11-23
>
> We are grateful for the reviewer's constructive feedback and insightful comments. In the following, we provide point-by-point responses to all your comments, and we hope our responses can help to resolve your concerns.
>
> **W1["In Figure 1-(a), MCP is compared only with MR and Oracle."]**
>
> We are grateful for the reviewer’s insightful comment. Following your suggestion, we have updated Figure 1(a) in the revised manuscript to include a comparison with APPO [1], in addition to MR [2] and Oracle. The new results show that, in the low-feedback regime, MCP consistently attains higher success rates than APPO while remaining competitive when more feedback is available. This empirical behavior is consistent with our theoretical analysis on the improved sample efficiency of MCP.
>
> **W2["The training curves are presented only for MCP."]**
>
> We appreciate the reviewer’s suggestion. Following your suggestion, we updated Figure 1(b) to include other baselines. From Figure 1(b), MCP attains convergence after roughly 50k training steps, whereas APPO requires around 70k steps to attain numerical convergence, and MR continues to improve until about 250k steps. This comparison highlights the faster convergence of MCP.
>
> **Q1["The authors mention that MCP provides an implicit way of encoding conservatism..."]**
>
> We thank the reviewer for this very helpful question. Below, we give an intuitive explanation of how the minimax structure in MCP encodes conservatism.
>
> From the preference dataset $\mathcal{D}$, we first construct confidence sets for the reward model and the dynamics at each stage $h$. The reward confidence set is as follows: $\mathcal{R}\_{\mathcal{D}}=\left \lbrace r\in \mathcal{R}:\sum\_{n=1}^N \log P\_r(o=o^n\mid\tau^{n,0},\tau^{n,1})\geq\sum\_{n=1}^N\log P\_{\hat{r}}(o=o^n\mid\tau^{n,0},\tau^{n,1})-\zeta \right  \rbrace, $
>
> and the transition confidence set at stage $h$ is: $\mathcal{P}^h\_{\mathcal{D}}=\left \lbrace P\_h\in\mathcal{P}\_{h}:\sum_{n=1}^N\sum\_{i=0}^1\log P\_h(s\_{h+1}^{n,i}\mid s\_h^{n,i},a\_h^{n,i}) \geq \sum\_{n=1}^N\sum\_{i=0}^1\log\hat{P}\_h(s\_{h+1}^{n,i}\mid s\_h^{n,i},a\_h^{n,i})-\zeta\_{P\_h}\right \rbrace.$
>
> Here $\hat r$ and $\hat P_h$ are the maximum–likelihood models, and $\zeta,\zeta_{P_h}$ control the size of the confidence sets. Intuitively, $\mathcal{R}\_{\mathcal{D}}$ and $\mathcal{P}^h_{\mathcal{D}}$ contain all models whose log-likelihood is close to that of $\hat r$ and $\hat{P}_ h$, i.e., all models that are statistically plausible given the data.
>
> Given these confidence sets, MCP chooses a policy by solving: $\hat{\pi}=\arg\max\_{\pi\in\Pi\_\mathrm{his}}\min\_{r\in\mathcal{R}\_\mathcal{D},\lbrace P\_h\in\mathcal{P}^h_\mathcal{D}\rbrace\_{h=0}^{H-1}} J(\pi;r,\lbrace P\_h\rbrace\_{h=0}^{H-1})-\mathbb{E}\_{\tau\sim\mu\_\mathrm{ref}}[r(\tau)].$
>
> For a fixed policy $\pi$, the inner minimization selects the worst-case reward and dynamics inside the confidence sets. The value $\min\_{r\in\mathcal{R}\_\mathcal{D},P\_h\in\mathcal{P}^h_\mathcal{D}}J(\pi;r,\lbrace P\_h\rbrace\_{h=0}^{H-1})-\mathbb{E}\_{\tau\sim\mu\_\mathrm{ref}}[r(\tau)]$ should be read as: how much improvement over the reference policy can we guarantee, even if the true model is the most pessimistic one that is still consistent with the data?
>
> Maximizing this worst-case improvement forces MCP to favor policies that:
> (1) perform well across all plausible models, and (2) avoid policies whose apparent gains are due to overly optimistic models in the confidence sets. In this sense, the minimax structure implicitly enforces conservatism: MCP always evaluates candidate policies under the most pessimistic model consistent with the data, rather than under an optimistic estimate. In theory, the confidence radii $\zeta$ and $\zeta_{P_h}$ depend on unknown problem-dependent quantities (e.g., concentrability coefficients). These constants cannot be computed reliably in practice, so we cannot form the exact confidence sets above.
>
> To obtain a practical algorithm, we move the confidence-set constraints into the objective via Lagrange multipliers, which leads to the regularized planning objective in MCP with weights $(\lambda_1,\lambda_2)$ on the reward and transition penalties. These regularization weights play the role of tunable surrogates for the intractable confidence radii: larger $\lambda$ corresponds to a larger effective confidence set and hence a more conservative plan.

---

> > ### Author Response · Authors · 2025-11-23
> >
> > **Q2["In Table 5, $\lambda_1$, $\lambda_2$ and $\lambda_3$ are important parameters for MCP."]**
> > We thank the reviewer for pointing this out. In the revised manuscript, we report the exact values of the three MCP hyperparameters $(\lambda_1, \lambda_2, \lambda_b)$, which control the regularization on the reward model, the transition model, and the behavior-cloning loss, respectively, for each dataset, as can be seen in the appendix (line 1861). In addition, we include a sensitivity analysis on the drawer-open and sweep-into tasks (see the tables below). As shown in the table below, the success rate of MCP changes only mildly when each hyperparameter is varied over several orders of magnitude, indicating that MCP is not highly sensitive to the precise choice of $(\lambda_1, \lambda_2, \lambda_b)$ and demonstrating the robustness of our method with respect to these hyperparameters.
> >
> >
> > drawer-open:
> > | $\lambda_1$ | 1e-1   | 3e-1       |     1       | 3      | 1e1 |
> > |------------|-------|------------|-------------|--------|--------|
> > |success rate|94.8±3.85|97.60±2.33|99.07±0.38|94.53±4.03|86.27±10.31|
> >
> > | $\lambda_2 $| 3e-3   | 1e-2       |     3e-2       |  1e-1     | 1 |
> > |------------|-------|------------|-------------|--------|--------|
> > | success rate|90.93±7.3|95.47±4.10|99.07±0.38|89.6±9.53|80.13±15.19|
> >
> > |$\lambda_b$ |  1e-1   | 3e-1       |     1       | 3      | 1e1 |
> > |------------|-------|------------|-------------|--------|--------|
> > | success rate|92±5.66|97.2±2.04|99.07±0.38|96.4±3.81|89.87±9.92|
> >
> >
> > sweep-into:
> > | $\lambda_1$ | 1e-1   | 3e-1       |     1       | 3      | 1e1 |
> > |------------|-------|------------|-------------|--------|--------|
> > |success rate|32.4±5.85|34.13±3.27|30.67±4.71|24±9.27|19.87±7.92|
> >
> > | $\lambda_2$ | 3e-3   | 1e-2       |     3e-2       |  1e-1     | 1 |
> > |------------|-------|------------|-------------|--------|--------|
> > | success rate|20.8±10.32|23.6±5.85|28.27±7.08|34.13±3.27|25.73±6.15|
> >
> > |$\lambda_b$ | 1e-1   | 3e-1       |     1       | 3      | 1e1 |
> > |------------|-------|------------|-------------|--------|--------|
> > | success rate|22.53±7.39|26.8±4.25|29.2±6.76|34.13±3.27|31.6±5.02|
> >
> >
> >
> > [1] Adversarial policy optimization for offline preference-based reinforcement learning (Kang et al., 2025)
> >
> > [2] Preference transformer: Modeling human preferences using transformers for RL (Kim et al., 2023)

---

> > > ### Comment · Reviewer_6XMk · 2025-11-27
> > > **Re: rebuttal**
> > >
> > > Thank the authors for their detailed rebuttal. The author's explanations adequately addressed my concerns.
> > >
> > > The revised version provides an intuitive explanation of how the minimax structure in MCP encodes conservatism. In addition, the learning curves in Figures 1-(a) and 1-(b) clearly demonstrate that MCP outperforms the other baselines.
> > >
> > > If the authors have more time to run additional experiments, the reviewer suggests including comparisons with MCP, APPO, MR, and Oracle on more datasets. Since MCP already outperforms these baselines on the BPT-wall, lever-pull, and drawer-open datasets in Table 2, the trends shown in Figures 1-(a) and 1-(b) are largely in line with what we would expect.
> > > Thus, it would also be informative to report how its performance scales with data size on other datasets (e.g., box-close, dial-turn) in the appendix, even if the performance is smaller there.
> > >
> > > Overall, the rebuttal satisfactorily addressed all comments. I will change my previous negative assessment to a positive one and raise the score to 6.

---

> > > > ### Author Response · Authors · 2025-11-27
> > > >
> > > > Thank you very much for your encouraging feedback, your recognition of our work, and your thoughtful suggestions. We are glad that the revised version addresses your concerns. We really appreciate your ideas about broadening the empirical study to better benchmark our work within the existing literature, and we will do our best to incorporate these directions in the next revision of the manuscript.

---

### Official Review · Reviewer_wXXH · 2025-11-01

**Soundness:** 3
**Presentation:** 3
**Contribution:** 3
**Rating:** 6
**Confidence:** 4

**Summary:**

This paper tackles incomplete data coverage and high computational cost in offline preference-based RL by introducing Model-based Conservative Planning (MCP). MCP uses a model-based planning framework that implicitly enforces conservatism, enabling sample-efficient and computationally tractable policy learning with general function approximation. Theoretically, MCP competes with the best policy supported by the dataset under partial coverage, with regret bounds improved via dynamic structures (e.g., kernelized nonlinear regulators, factorized models). Empirically, across 8 Meta-World tasks, MCP outperforms baselines like APPO and IPL in average rank and shows strong robustness in low-data regimes.

**Strengths:**

1. simultaneously achieve sample efficiency and computational tractability in offline PbRL with unknown dynamics and partial coverage. It encodes conservatism via relative performance instead of explicit confidence sets or extra value modeling, simplifying the framework and overcoming the computational bottlenecks of methods like FREEHAND and Sim-OPRL.

2. Supports general function approximation (linear models, neural networks, etc.), and derives adaptive concentration coefficients and regret bounds for structured dynamics (e.g., kernelized nonlinear regulators and factorized models), mitigating the curse of dimensionality and broadening theoretical applicability.

**Weaknesses:**

1. Compared with the APPO algorithm, this paper mainly introduces a model with conservative terms, but the experimental results do not show much improvement, while introducing additional model computational overhead.

2. What is the robustness of MCP when there is noise in preference labels? Quantitative analysis of the impact of label noise on performance can be conducted to prove that the design of MCP can have better generalization.

3. Each time a policy is updated, a new MDP model should be learned to challenge it, which is very wasteful in terms of compuational cost.

**Questions:**

see above.

---

> ### Author Response · Authors · 2025-11-23
>
> We are grateful for the reviewer's constructive feedback and insightful comments. In the following, we provide point-by-point responses to all your comments.
>
> **W1["Compared with the APPO [1] algorithm, this paper mainly introduces a model with conservative terms, but the experimental results do not show much improvement..."]**
>
>
> Thank you for this comment. Our goal is not only to achieve a modest empirical gain over APPO, but to develop a different conservative PbRL method that is both more sample-efficient and theoretically simpler.
>
> Conceptually, MCP and APPO take very different routes to conservatism. APPO is a pessimistic **value-based** algorithm: it learns a value/Q-function and then adds a pessimism term on top of that value estimate. This requires (i) specifying a realizable value-function class and (ii) running double Q-learning with target networks and other tricks to stabilize value training, which increases computational burden. In contrast, MCP is a **model-based planner**: we learn a reward model and a dynamics model, and impose conservatism directly in the planning objective via the relative-performance term. As a result, MCP does not need any value-function estimation, and the only function classes we assume are those for the reward and transition models. This difference also leads to distinct theoretical guarantees. We improve the PAC sample complexity from APPO’s $O(\epsilon^{-4})$ to $O(\epsilon^{-2})$, where $\epsilon$ is the suboptimality gap.
>
> Empirically, MCP has the following advantages: (i) **overall performance**: although APPO is already a strong state-of-the-art baseline, MCP still achieves the best average rank across all 8 Meta-World tasks in Table 2; (ii) **sample efficiency**: MCP outperforms APPO in the low-data regime (100–500 preferences) in the new Fig. 1(a) which aligns with our theoretical analysis; (iii) **training behavior**: the training curve (new Fig. 1(b) shows that MCP converges faster than APPO; and (iv) **computational cost**: our runtime measurements (Table 1 below) show that MCP has slightly lower computational cost than APPO for 100k training steps.
>
>
>
> **W2["What is the robustness of MCP when there is noise in preference labels?"]**
>
> Thank you for raising the question about robustness to noisy preference labels. We evaluate MCP and MR [2] under synthetic label noise by introducing label flipping: for each pairwise preference, we flip the label with probability $p$, and vary $p$ across several noise levels. In the revised manuscript, we have added Figure 2(b), which presents the success rates on the drawer-open task under varying label-flipping probabilities. As the probability of label flipping increases, the performance of both methods degrades, but MCP consistently maintains higher success rates and exhibits a slower decay compared to MR, especially at larger noise levels. This empirical result indicates that the conservative design of MCP yields improved robustness and hence better generalisation in the presence of noisy preference labels.
>
> **W3["Each time a policy is updated, a new MDP model should be learned to challenge it, which is very wasteful in terms of compuational cost."]**
>
> Thank you very much for this insightful point. In our implementation, we do not relearn a new dynamics and reward model from scratch after every policy update. Instead, we keep the existing model and only refine it across iterations. Specifically, at iteration $t$, we initialize the optimization with the parameters from the previous iteration, i.e., we warm-start from $(r_{t-1}, P_{t-1})$ when solving the conservative objective. This warm-start strategy substantially reduces the additional computational cost.
>
> We also compare the computational cost of MCP with the baselines. We measure the time needed to run 100k training steps on the same machine with a single NVIDIA RTX 5090 GPU. As shown in Table 1 below, MCP requires slightly less time than APPO while achieving comparable performance, and runs somewhat slower than MR but with significantly higher performance. To further understand where the cost comes from, Table 2 decomposes the MCP runtime into the time spent on step 5 (optimizing the conservative objective with the learned model) and step 6 (policy update). The sum of these two parts matches the total MCP time, showing that the overall computational complexity of MCP remains practical.
>
> Table 1
> | Method |MR|APPO|MCP|
> |--------|------------------|-------------------|-------------------|
> | Total time (min) |22.7|36.8|33.5|
>
> Table 2
> | Method |MCP-step5|MCP-step6|MCP|
> |--------|------------------|-------------------|------------------|
> | Total time (min) |24.2|9.3|33.5|
>
>
> [1] Adversarial policy optimization for offline preference-based reinforcement learning (Kang et al., 2025)
>
> [2] Preference transformer: Modeling human preferences using transformers for RL (Kim et al., 2023)

---

### Author Response · Authors · 2025-11-23

We would like to thank all reviewers for their thoughtful and constructive feedback. We are pleased that Reviewers wXXH, 6XMk, GUzu, and Ec41 view our theoretical framework and guarantees as sound and technically solid. We also appreciate that Reviewer Ec41 further emphasizes the novelty of our conservative planning setup for PbRL, the importance of the problem, and the support provided by our simulation results..

Following the reviewers' comments, we have revised the manuscript accordingly and uploaded an updated version (rebuttal PDF), in which all changes are highlighted in blue. In this global response, we summarize all our main revisions of our paper and then provide a point-by-point discussion in the individual response to each reviewer for all their feedback.

**Ablation studies on the main components of our algorithm.**  We added an ablation study (new Fig. 2(a)) that isolates the effects of reward regularization, transition regularization, the relative-performance term, and the model-based planner. Full MCP rapidly reaches a high and stable success rate, whereas removing either regularizer leads to little improvement, dropping the relative-performance term results in highly variable and unstable learning, and replacing the planner with a value-based variant improves only in the early stage and then degrades. These results indicate that all four conservative components are necessary for the strong and stable performance of MCP. We refer the reviewers to the revised manuscript for a more detailed discussion (lines 477-489).

**Ablation study on model robustness to noise.** We also added a label-noise experiment, shown in the new Figure 2(b). As the label-flipping probability increases, MCP maintains higher success rates and degrades more slowly than MR, indicating better robustness to noisy preference labels. We refer the reviewers to the revised manuscript for a more detailed discussion (lines 503-508).

**Sensitivity analysis.** We perform a hyperparameter sensitivity study for the regularization weights $\lambda_1$ and $\lambda_2$ on the drawer-open-1000 and sweep-into-1000 tasks. In Table 3, we vary each coefficient over a wide range (roughly from $[10^{-3}, 10^{1}]$) while keeping the other weights fixed, and report the mean success rate over three seeds. The performance of MCP changes only mildly across this range—for example, on drawer-open-1000, MCP attains a peak success rate of about $99%$, and most other choices of $\lambda\_1$ and $\lambda_2$ keep the performance within roughly $5$–$10$ percentage points of this peak, showing that our method is not overly sensitive to precise tuning. Together, these results indicate that MCP is reasonably robust to the choice of regularization weights. Moreover, thanks to the safe policy improvement guarantee (Corollary 4.2), practitioners can safely tune $\lambda\_1$ and $\lambda_2$  online by trying multiple settings and selecting the best-performing policy, while still ensuring that the resulting policy is essentially no worse than the behavior policy. For a more detailed analysis, we refer the reviewers to our revised manuscript (lines 520-523).


**Experiments on high-dimensional PbRL tasks.**  In the main paper, we evaluate MCP on Meta-World tasks, where the state and action dimensions are at most around \(39\) and \(4\), respectively. In this rebuttal, we additionally construct two higher-dimensional offline PbRL benchmarks based on MuJoCo Ant (state dimension \(105\), action dimension \(8\)) and Humanoid (state dimension \(348\), action dimension \(17\)). As summarized in Table 8, MCP achieves the best average return (over three seeds) among the compared baselines on both tasks, showing that our method continues to perform well in substantially higher-dimensional settings. For a more detailed analysis, we refer the reviewers to our revised manuscript (lines 1944-1953).

**Computational complexity.** We empirically measure the runtime of MCP and compare it with MR and APPO on the same task and hardware (Tables 9 and 10). The results show that MCP runs slightly faster than APPO and has a computational cost comparable to MR, while achieving substantially higher performance than MR. This indicates that the additional model-based planning in MCP does not introduce a significant overhead in practice. For a more detailed analysis, we refer the reviewers to our revised manuscript (lines 1962-1965).


We once again thank all reviewers for their thoughtful comments and constructive feedback, which have greatly helped us strengthen the paper. We would be happy to provide any additional clarification that may be useful for your final evaluation.

---

### Meta-Review · Area_Chair_VaX9 · 2026-01-03

**Summary:**

This work proposes Model-based Conservative Planning, an algorithm for offline preference-based reinforcement learning. The authors go beyond other works in the area by showing their algorithm has guarantees for scenarios for partial coverage and allows for a tractable implementation. Their guarantees improve over Zhu 2023, and Kang & Oh, 2025 by providing guarantees for factored models and for kernelized non-linear regulators. The authors show the tractability and usability of their algorithms in meta-world domains. The less enthusiastic reviewers were satisfied with the author’s rebuttal and expressed desire to raise their score.

**Reviewer Concerns:**

The main concern of reviewers was the limited scope of the experiments and producing additional ablation studies. These were mostly addressed by the authors during the rebuttal, as it was acknowledged by the reviewers.

**Reviewer Scores:**

One of the reviewers expressed their desire to raise their score from a 4 to a 6. The other reviewers had a positive assessment of the work.

---

### Decision · Program_Chairs · 2026-01-26

Accept (Poster)